# Comprehensive analysis of locomotion dynamics in the protochordate *Ciona intestinalis* reveals how neuromodulators flexibly shape its behavioral repertoire

**Athira Athira**©, **Daniel Dondorp**©, **Jerneja Rudolf**¤©, **Olivia Peytral**, **Marios Chatzigeorgiou** *

Sars International Centre for Marine Molecular Biology, University of Bergen, Bergen, Norway

© These authors contributed equally to this work.
¤ Current address: Kavli Institute for Systems Neuroscience, NTNU,Trondheim, Norway
* Marios.Chatzigeorgiou@uib.no

**Data Availability Statement:** We have established a repository containing datasets corresponding to this study. Specifically, it contains 1. Multi-point

## Abstract

Vertebrate nervous systems can generate a remarkable diversity of behaviors. However, our understanding of how behaviors may have evolved in the chordate lineage is limited by the lack of neuroethological studies leveraging our closest invertebrate relatives. Here, we combine high-throughput video acquisition with pharmacological perturbations of bioamine signaling to systematically reveal the global structure of the motor behavioral repertoire in the *Ciona intestinalis* larvae. Most of *Ciona*'s postural variance can be captured by 6 basic shapes, which we term "eigencionas." Motif analysis of postural time series revealed numerous stereotyped behavioral maneuvers including "startle-like" and "beat-and-glide." Employing computational modeling of swimming dynamics and spatiotemporal embedding of postural features revealed that behavioral differences are generated at the levels of motor modules and the transitions between, which may in part be modulated by bioamines. Finally, we show that flexible motor module usage gives rise to diverse behaviors in response to different light stimuli.

## Introduction

A primary function of animal nervous system is to transform sensory input into a sequence of actions known as behavioral output. Thus, the overarching motive of neurobiology research is to delineate the functional makeup and mechanistic basis of these behavioral outputs. Major progress has relied on the development of experimental tools and analysis methods that permit real time measurements and quantitative characterization of behavior (reviewed in [1–5]). Among the various natural animal behaviors, locomotion forms an integral part of nervous system function. Researchers in the field have been able to employ the aforementioned modern technologies to define motor actions as a function of their natural stereotyped elements, known as behavioral "modules," "motifs," "syllables," or "primitives" [1,6–10], where these

tracking data of the larvae obtained using the Tierpsy Tracker (skeletons) 2. Features like curvature, speed, etc calculated from the tracking data 3. The results of time-series analyses approaches (matrix profiling, HMM, Spatio-temporal clustering) performed on the feature dataset. 4. The Hidden Markov Models trained for inferences The link to the zenodo repository is: https://zenodo.org/record/6761772#.YrsDNexBxyF The DOI for these datasets is: 10.5281/zenodo.6761772 Code for acquisition software can be found here: https://github.com/ChatzigeorgiouGroup/imMobilize Notebooks for Matrix profiling analysis is available here: https://github.com/ChatzigeorgiouGroup/ciona_behaviour_matrix_profile Code for biophysical features, HMM, spatio-temporal embedding and statistical analysis: https://github.com/ChatzigeorgiouGroup/behavior_ciona_bioamines.

**Funding:** This project has been funded by a grant of the Research Council of Norway, of which M.C. is the PI: grant number 234817 (Sars International Centre for Marine Molecular Biology Research, 2013-2022). URL: https://www.forskningsradet.no/ "The funders had no role in study design, data collection and analysis, decision to publish, or preparation of the manuscript."

**Competing interests:** "The authors have declared that no competing interests exist."

**Abbreviations:** ASW, artificial seawater; CW, clockwise; CCW, counter-clockwise; DBSCAN, density-based spatial clustering of applications with noise; ddN, descending decussating neuron; EC, Eigenciona; EM, expectation maximization; G-HMM, Gaussian hidden Markov model; HMM, hidden Markov model; PCA, principal component analysis; SMD, standardized median difference; SSRI, serotonin reuptake inhibitor; TPH, tryptophan hydroxylase; tSNE, t-distributed stochastic embedding.

basic building blocks of motor behavior operate under organizational and hierarchical rules that bear similarities to phonological and syntactical rules that govern language. Modern systems neuroscience approaches have greatly facilitated the investigation of vertebrate motor modules [11–13], which in invertebrates are even more likely to be interrogated with high sensitivity and precision, largely due to the latter's smaller nervous systems [14–16]. In addition, the next generation of neuroscience discovery capitalizes on developing and studying new nontraditional model species to reveal not only common principles, but also differences in behavioral organization across the tree of life as well as within important clades [17–19]. Consequently, there is an urgent requirement for expanding neuroethological studies to additional organisms occupying key phylogenetic positions.

Invertebrate chordates belonging to the phylum Chordata are obvious candidates for neuroethological analysis since they are close relatives of vertebrates and may provide important insight into the evolution of chordate nervous systems. While the importance of studying invertebrate chordates has been recognized in the field of evo-devo, as evident from an explosion of evolutionary, genomic, and developmental studies primarily in 3 organisms: the cephalochordate amphioxus and the tunicates *Ciona intestinalis* and *Oikopleura dioica*, these organisms have yet to be leveraged in the context of neuroscience. Understanding their nervous system functions and behavioral repertoire will provide insights into the conservation and diversity of locomotory circuits and how these relate to the evolution of the diverse modes of locomotor behavior [20].

Recent publication of the *C. intestinalis* larval connectome [21,22], single-cell transcriptomes of the larval nervous system [23,24] and establishment of in vivo functional imaging [25,26] have made *Ciona* a promising target for functionally dissecting a small invertebrate nervous system at a systems level.

However, a major hindrance to employing *Ciona* larvae for systems neuroscience is the absence of a behavioral platform that can measure phenotypes in an extensive and intensive manner [27], which is especially crucial for the analysis of locomotion due to its sensitivity to both neurogenetic [28,29] and neuropharmacological perturbations [30–34].

In this work, we address this knowledge gap by using machine vision to track, skeletonize, and extract postural features from thousands of larvae swimming both spontaneously and under light stimulation. We additionally combine wild-type swimming behavioral analysis with a small-scale pharmacobehavioral screen that targets bioamine signaling, a key regulator of the biophysical properties of neurons, synapses [30,35,36], and behavior [30,37]. Using dimensionality reduction, we derive lower dimensional representations of body postures, which we term "eigencionas." With these, we can explain the majority of postural variance in the *Ciona* larvae. We also combine 3 state-of-the-art approaches: motif identification, hidden Markov model (HMM), and spatiotemporal embedding to quantitatively define *Ciona* larval behavioral dynamics and thereby uncover the perturbation-sensitive modulation effects exerted on them by bioamine neuromodulators.

## Results

### Parametrization of *Ciona* using interpretable features

Using 5 inexpensive USB microscope-based tracking setups (Figs 1A and S1A–S1P), we recorded high-resolution videos of 1,463 individually and freely swimming *C. intestinalis* larvae of which 694 were wild-type larvae swimming in artificial seawater (ASW) and 769 were incubated with drugs that target different neuromodulators (S1 and S2 Tables).

We then utilized the Tierpsy software [38,39] to extract, in a high-throughput manner, 49 two-dimensional positional coordinates of each of the contours and midlines of each *Ciona*

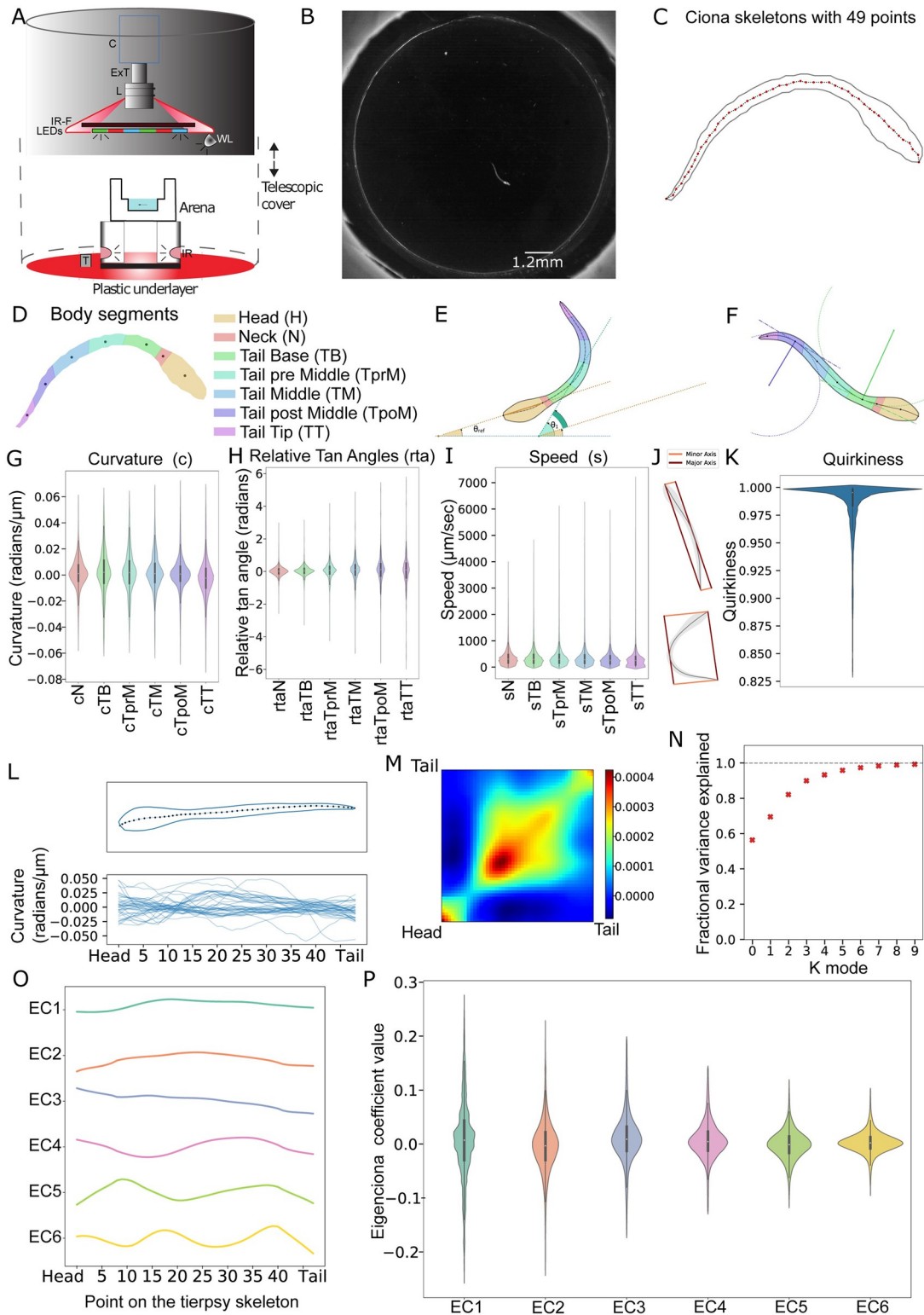

**Fig 1. Characterization of *C. intestinalis* swimming using core features and eigencionas.** (A) Schematic of the Ciona Tracker 2.0 video acquisition setup. Each setup was composed of a monochrome camera (C), connected to an extension tube (ExT) and a magnification lens (L). A holder piece housed an IR filter (IR-F), a set of LEDs and a white light source (WL). The arena was nested in a PLA ring that housed the infrared lights used for dark field illumination (IR). A plastic underlayer secures the PLA ring and a thermometer that reports temperature for each acquisition setup. A telescopic plastic cover shields the arena from

external light sources. (B) An image of the raw data (before processing with Tierpsy) showing a larva swimming in the arena. The average length of a *Ciona* was 1,330.61 μm or equivalently 115.10 pixels. (C) *Ciona* skeletons generated using Tierpsy are defined by 49 equally spaced points on their midlines. (D) Skeletons were divided into 7 body segments along the length of the animal. (E) Definition of angles: θ1 refers to the tangent angle of a particular segment, θref refers to the tangent angle for the head segment (with respect to the horizontal axis). The difference θ1-θref is defined as the relative tangent angle for each segment. (F) Curvature definition visualization. Curvature is defined as inversely proportional to the radius of the osculating circle at a given point on the skeleton. Green shows lower curvature and blue shows the centroid of a segment with a higher curvature. (G–I) Distribution of curvature, relative tangent angle, and speed values for each body segment in wild-type larvae ($n = 694$ larvae). (J) Quirkiness is defined as the ratio of the major and minor axes of the body as illustrated. (K) Distribution of the quirkiness for wild-type larvae ($n = 694$ larvae). (L) A skeleton with the contour and 49 points are shown as an example. Curvature values along the 49 points for a set of randomly sampled skeletons show variation in the skeleton postures. (M) Covariance matrix calculated from curvature values of a subset of wild-type skeletons. The smooth structure of the covariance matrix indicates that postures can be represented using a small number of eigencionas ($n = 231$ larvae). (N) Six eigencionas are sufficient to explain 97% of the variance in the curvature. (O) Visualization of the top 6 eigencionas obtained by an eigen decomposition of the covariance matrix, shown in descending order of the fraction of the variance explained. X-axis refers to points along the skeleton. (P) Distribution of eigencoefficient values for wild-type larvae ($n = 694$ larvae). For statistical analysis, we first tested for data normality using Shapiro–Wilk test ($\alpha = 0.05$). To compare between different body segments, we used the Wilcoxon signed-rank test ($\alpha = 0.05$) (see S3–S6 Tables for the underlying data).

larva, from our videos. This allowed us to approximate all the larvae via the 49 positional coordinates of their midlines (skeletons) during our downstream analysis (Fig 1B and 1C).

Next, we grouped the 49 points identified by Tierpsy into 7 distinct segments ranging from head to tail (Fig 1D and see Methods). We are more interested in the 6 segments from neck (N) to tail tip (TT), given that we found that the head segment is rigid (S2A–S2C Fig). We then defined relative tangent angles for each of these 6 segments relative to the head segment (Fig 1E). The relative tangent angles provide a measure of the orientation of each of the segments with respect to the head segment. Curvature values, on the other hand, give a quantitative measure of the local bend at the middle of each of the segments, which is independent of the overall shape of the larva (Fig 1F). This difference can be seen in the violin plots of Fig 1, where the wild-type larvae curvature in each of the segments have a similar range of values (Fig 1G), whereas for the relative tangent angles, the range (variance or spread) of values become wider as we move away from the neck segment (Fig 1H). The highest mean segment speeds are seen in the neck region (sN), while the lowest mean speeds are observed at the tail post middle (sTpoM) and tail tip (sTT) segments (Fig 1I).

We also calculated quirkiness values that give a related measure of eccentricity. A quirkiness value of 1 would mean that the skeleton has a perfect straight-line shape and quirkiness values closer to zero would indicate a skeleton where the bounding box (Fig 1J) is nearly a square that encloses a highly curved animal. The quirkiness distribution of the wild-type dataset is in line with our empirical observation that the *Ciona* larvae, while stationary or swimming, primarily maintain a relatively straight body posture where exaggerated tail bends are rare. These are reflected in the lower tail of the violin plot (Fig 1K).

While features like curvature and relative tangent angles could describe postures very accurately, the richness in these descriptions comes at a cost of very high dimensionality. Our aim was to obtain a simpler representation that describes the wide range of postures that *Ciona* can obtain as represented by a randomly sampled set of curvature values (Fig 1L) without losing significant information [40,41]. We initially examined if there are any dependencies or relationships between the 49 points on the skeleton by looking at the covariance matrix of curvature values from 231 experiments, amounting to over $2 \times 10^6$ images or skeletons (Fig 1M). As expected, this matrix indicates a strong correlation between adjacent points, indicating that the 49 points do not move independently of each other. We confirmed this by performing principal component analysis (PCA) in the form of an eigenvalue decomposition of the covariance matrix [41] and found that 97% of the variance observed in the curvature data can be

explained by 6 eigenvectors (here after termed eigencionas) (Fig 1N). For any given frame in the video, the curvatures of the skeleton can be approximated as a linear combination of these 6 eigencionas: EC1 to EC6 (Fig 1O). Hence, the coefficients (here after termed eigencoefficients) of these 6 eigencionas were used for further analysis as a simpler but nearly accurate description of skeleton postures (Fig 1P).

Following the calculation of the biophysical features as described above, we looked at the statistical differences in each of them across the wild type and drug-treated experimental groups. We employed different metrics and visualizations for this purpose. First, we used a summary statistic called standardized median difference (SMD) [42] to compare the distributions of features obtained from the different drug datasets with our wild-type dataset (Fig 2A). The SMD values demonstrate that for the segments' speeds the largest increase is observed upon treatment with phentolamine, which is an α-adrenergic antagonist and raclopride, a $D_2$ dopamine receptor antagonist, while the largest decrease is observed in animals treated with imipramine, a potent serotonin reuptake inhibitor (Fig 2A and S7 Table). Our findings using raclopride are consistent with pervious observations we made in Rudolf and colleagues, where the dopamine transporter inhibitor modafinil decreased larval swimming speed [43]. For quirkiness, chlorpromazine, an antipsychotic drug, has the lowest SMD value, while quinpirole, a selective $D_2$/$D_3$ receptor antagonist, exhibits the highest SMD value. Multiple drugs showed an overall increase in body curvature and relative tangent angles across most body segments. Paroxetine, a serotonin reuptake inhibitor (SSRI) and phentolamine showed the biggest increase relative to wild type. In contrast, both raclopride and quinpirole showed substantially decreased values for these features (Fig 2A and S7 Table). These trends are also observed in the visualizations using split violin plots of the distributions (S3–S5 and S6A Figs).

Then, to examine the differences in the eigencoefficient features, we present a bubble grid chart (Fig 2B) where the hue and radius of the circles respectively represents the mean and the standard deviation of the distribution. Drug treatments that resulted in statistically significant changes in EC values are summarized in S13–S15 and S53 Tables. For EC1, phentolamine and paroxetine have a significantly larger positive mean for EC1, suggesting that the EC1 component largely represents the shapes observed in larvae treated with these drugs (S6B Fig and S13–S15 and S53 Tables). Conversely, for EC2, fluoxetine has a significantly larger negative mean with high standard deviation (S6B Fig and S13–S15 and S53 Tables). Like EC1 and in contrast to EC2, EC3 exhibits a very strong positive trend across most drug treatments (S6B Fig and S13–S15 and S53 Tables) except for α-methyl serotonin and mianserin. Notably, the largest effects on the EC3 component contribution to skeleton postures were observed in larvae treated with imipramine and raclopride (S6B Fig and S13–S15 and S53 Tables). The remaining eigencoefficients, EC4 to EC6, show comparatively modest changes in mean values compared to wild type, except for EC5 in larvae treated with the SSRIs fluoxetine and paroxetine that exhibit a strong reduction in the mean value of the distribution. These trends were also noted in the visualizations using split violin plots of the distributions (S6B Fig). Overall, eigencoefficients are good descriptors of skeleton posture, and at least EC1 to EC3 are shown to be strongly up-regulated and down-regulated in our pharmacobehavioral screen.

## *Ciona* locomotion is rich in behavioral motifs across timescales

To measure the stereotypy and reveal the modular structure in the motor repertoire of *Ciona*, we used a multitude of state-of-the-art analytical methods including motif discovery, HMM, and spatiotemporal embedding into a lower dimensional space. The first approach we took was that of recurring motif discovery [9,44]. Recurring motifs indicate that some information is conserved for a system to produce the same output at least twice, in our case, a repeated

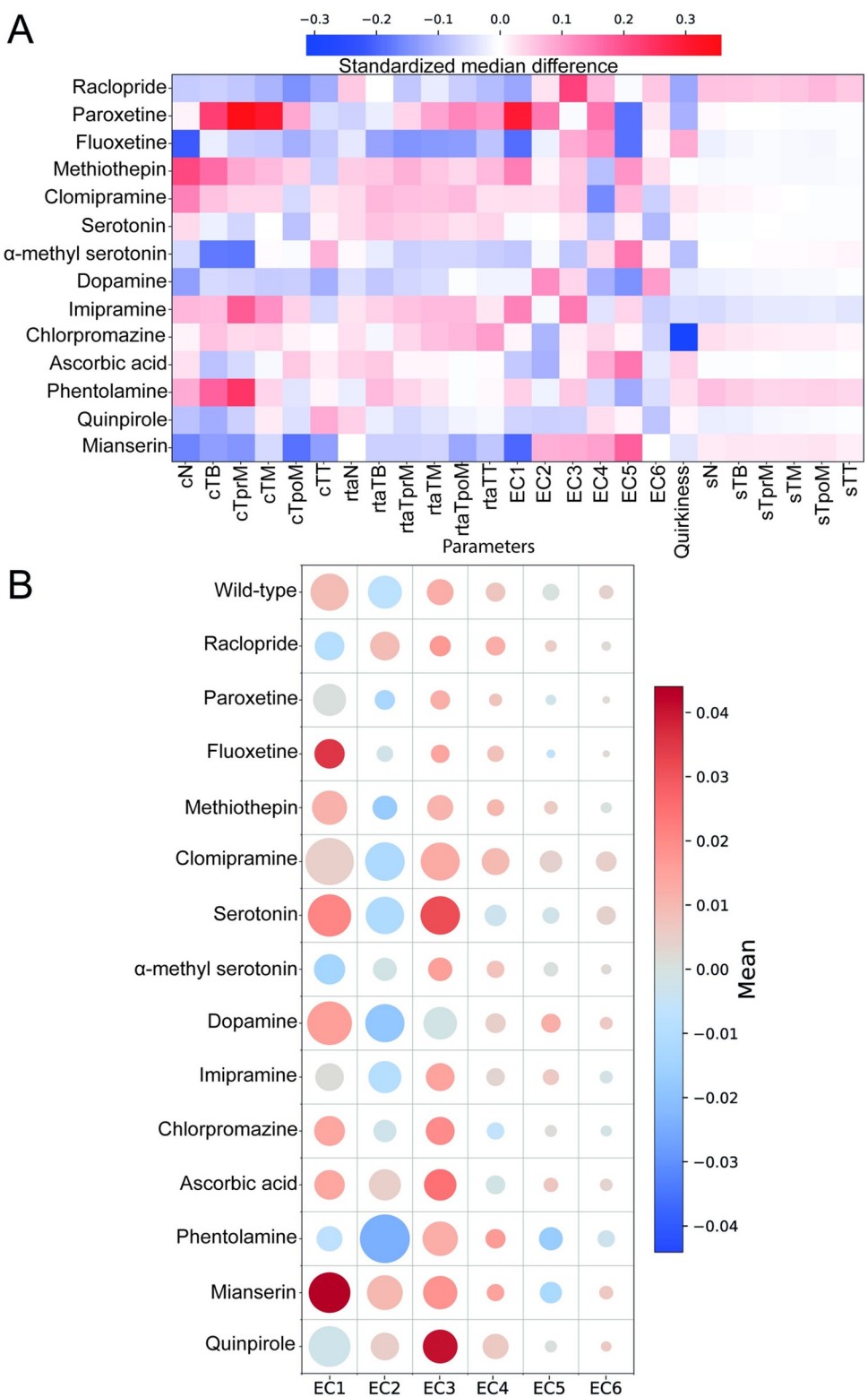

**Fig 2. Behavioral barcoding reveals the diverse effects of bioamines in locomotion features.** (A) Heatmap showing SMD of 25 features, calculated for drug-treated larvae relative to wild-type larvae. Working concentrations for all drugs are shown in Table 1. Note that SMD values for dopamine are calculated relative to an ascorbic acid solution that was used as a solvent for dopamine (SMD values shown in S7 Table). (B) Bubble grid chart showing effects of drug treatments in the use of eigencionas by swimming larvae, as quantified by the eigencoefficients. Color of circles

indicates the mean value of the eigencoefficient features as indicated by the colorbar, while their radius indicates the variance (please see S8 and S9 Tables and https://doi.org/10.5281/zenodo.6761771 for the underlying data). The number of animals and video frames contributing to this figure are indicated in S1 and S2 Tables. cN, curvature Neck; cTB, curvature Tail Base; cTM, curvature Tail Middle; cTpoM, curvature Tail post Middle; cTprM, curvature Tail pre Middle; cTT, curvature Tail Tip; EC1, Eigenciona 1; EC2, Eigenciona 2; EC3, Eigenciona 3; EC4, Eigenciona 4; EC5, Eigenciona 5; EC6, Eigenciona 6; rtaN, relative tan angle Neck; rtaTB, relative tan angle Tail Base; rtaTM, relative tan angle Tail Middle; rtaTpoM, relative tan angle Tail post Middle; rtaTprM, relative tan angle Tail pre Middle; rtaTT, relative tan angle Tail Tip; SMD; standardized median difference; sN, speed Neck; sTB, speed Tail Base; sTM, speed Tail Middle; sTpoM, speed Tail post Middle; sTprM, speed Tail pre Middle; sTT, speed Tail Tip.

behavioral action or state. To perform automated behavioral motif detection, we decided to use matrix profile, a computational tool that makes it possible to solve the dual problem of motif discovery and anomaly detection in a time series dataset [45–47]. The main advantages of matrix profiling are that it is robust, scalable, computationally efficient, and largely parameter free. For our analysis, we have employed the curvatures of the 7 body segment midpoints of the *Ciona* larvae. This results in 7-dimensional time series that we used to calculate matrix profiles and search for recurring motifs over 2 time windows: 1 second (30 frames) and 5 seconds (150 frames) to capture both short (spontaneous) and long (sustained) behaviors that repeat over time (Fig 3A). We identified the motifs over the 2 time windows that resulted in 2 datasets consisting of a set of 87,569 motifs over 1 second and a set of 18,776 motifs over 5 seconds. In the 1-second time window, we find repeating motifs that correspond to larvae performing different swimming maneuvers including clockwise (CW) or counter-clockwise (CCW) turns, straight runs, twitching, rapid accelerations, decelerations, and beat-and-glide, as well as startle-like escape actions (Fig 3B). In the 5-second time window, we found a lot of CW and CCW spiral swimming, circle swimming, straight runs that conclude in different ways: rapid halt, swim in a small circle, or perform a spiral swim. Other motifs include drifting, persistent unidirectional tail flicking, and again startle-like escape actions that take place over a longer time window (Fig 3C and S1 and S2 Movies).

We then asked whether there are differences between these motifs. Given that our 2 motif datasets are practically speaking large sets of short 7-dimensional time series, we performed time series clustering using k-means clustering (TimeseriesKMeans) to identify major groups to classify our motifs into. We determined the optimal number of clusters to be 15 for each of

**Table 1. Working concentrations of drugs used in this study.**

| Drug | Concentration [μm] | |
|---|---|---|
| Chlorpromazine hydrochloride | 0.1 | C8138-5G (Sigma-Aldrich) |
| S(−)-Raclopride (+)-tartrate salt | 0.025 | R121-25MG (Sigma-Aldrich) |
| Methiothepin mesylate salt | 1 | M149-100MG (Sigma-Aldrich) |
| Mianserin hydrochloride | 0.05 | M2525-100MG (Sigma-Aldrich) |
| Fluoxetine hydrochloride | 1 | F132-10MG (Sigma-Aldrich) |
| Clompiramine hydrochloride | 10 | C7291-5G (Sigma-Aldrich) |
| Imipramine hydrochloride | 10 | I7379-5G (Sigma-Aldrich) |
| Phentolamine hydrochloride | 50 | P7547-100MG (Sigma-Aldrich) |
| Quinpirole hydrochloride | 0.03 | Q102-10MG (Sigma-Aldrich) |
| Serotonin creatine sulfate | 100 | H7752-5G (Sigma-Aldrich) |
| α-Methylserotonin maleate salt | 5 | M110-10MG (Sigma-Aldrich) |
| Paroxetine hydrochloride | 1 | P9623-10MG (Sigma-Aldrich) |
| Dopamine hydrochloride | 0.1 | H8502-5G (Sigma-Aldrich) |
| L(+)-Ascorbic acid | 28 | 20150.184-100G (VWR) |

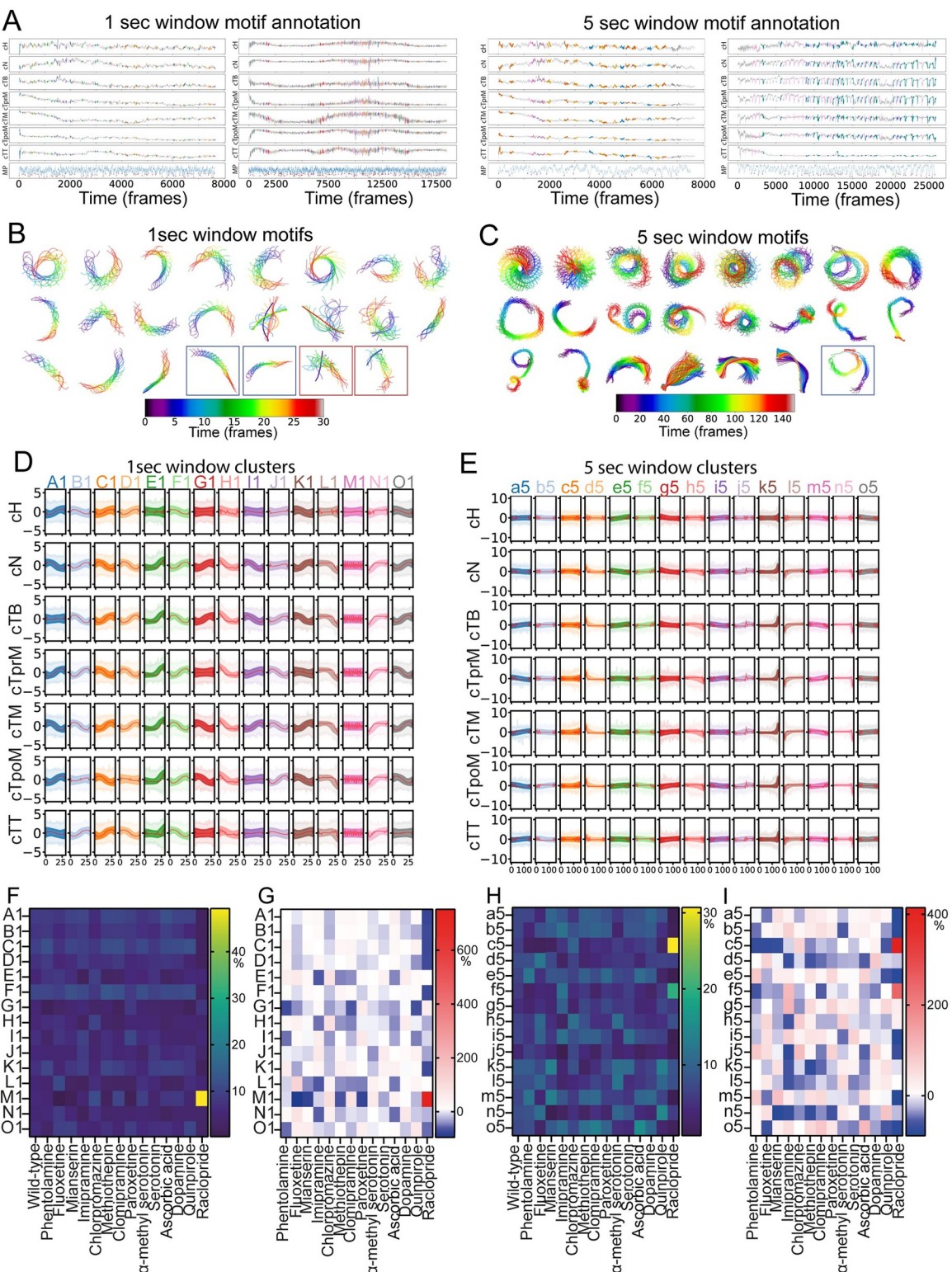

**Fig 3. Matrix profiling reveals a wealth of stereotyped behavioral motifs.** (A) Four representative cases of larval swimming analyzed using matrix profiling. Curvatures of 7 body segments were used as the input. The matrix profile for each animal is shown as an additional row at the bottom of each plot (labeled as MP). Each dip marked with a red dot indicates the onset of a motif that recurs in the dataset. Each motif is color coded according to the cluster it belongs to. (B) Representative examples of motifs that are enriched in the 1-second time window. Color-coded skeletons of the animals plotted in sequence to show time progression (0➔30 frames, i.e.,

blue→red). Enclosed in the blue boxes are example beat-and-glide motifs and in red boxes startle response motifs. (C) Representative examples of motifs that are enriched in the 5-second time window. Color-coded skeletons of the animals plotted in sequence to show temporal progression (0→150 frames, i.e., blue→red). Enclosed in a blue box is beat-and-glide motif. (D) Time series clustering of motifs for 1-second time window and (E) for 5-second time window. In the graphs, each column corresponds to 1 motif cluster, while each row corresponds to 1 of the 7 body segment curvatures. Each cluster was assigned a color and a letter. Red line in all clusters indicates cluster center, and the variance is shown as shaded lines. Use of capital letters in (D) and small letters in (E) is done to indicate that these are not the same clusters. (F) Heatmap visualization of the 1-second time window motif clusters representation (values in % can be found in S16 Table). (G) Heatmap visualization of the percentage fold changes relative to wild type for the data shown in panel F(values in % can be found in S17 Table). (H) Heatmap visualization of the 5-second time window motif clusters for different drugs (values in % can be found in S18 Table). (I) Heatmap visualization of the percentage fold changes relative to wild type for the data shown in panel H (values in % can be found in S19 Table). Drugs that resulted in a statistically significant up-regulation or down-regulation of the usage of 1- and 5-second motif clusters are listed in S54 and S55 Tables, respectively (please see S41–S46 Tables for the underlying statistical data). Dopamine values are compared relative to ascorbic acid and not with wild type. The underlying data, including all individual observations are available to download from: https://doi.org/10.5281/zenodo.6761771.

the time windows (Fig 3C and 3D) using the Elbow method, where we find the maximum of the second derivative of the curve showing variance explained over cluster numbers. From these clusters, the following involved actively swimming larvae (1sec: "H1," "L1," "M1," N1;" 5sec: "c5," "d5," "j5," "l5"), while other clusters (1sec: "A1," "D1," "E1," "G1;" 5sec: "f5," "g5," "h5," "k5," "n5") represent moderately active and (1sec: "B1," "C1," "F1," "I1," "J1," "K1," "O1;" 5sec: "a5," "b5," "e5," "I5," "m5," "o5") represent dwelling larvae. Random sampling of single-frame skeletons as well as 1- and 5-second long sequences of skeletons revealed behaviors that set apart the different clusters (Figs 3D, 3E and S7A–S7D). For example, in the active clusters, "M1," we find motifs corresponding to larvae performing sharp CW or CCW turns with high curvature tail beats, while clusters "L1" and "N1" are enriched in larvae showing startle-like behaviors and unidirectional tail flicking characterized by asymmetric tail beats. We were also able to identify biologically interesting clusters within the 5-second time window, such as active cluster "c5," which is enriched in CW and CCW circular and spiral swimming motifs as well as straight runs concluding with a circular or spiral maneuver (S7D Fig). Cluster "d5" is enriched in short range swimming and late onset escape maneuvers (S7D Fig). Interestingly, while cluster "l5" was enriched in motifs where most of the swimming activity occurred in the first 3 seconds (S7D Fig), clusters "k5" and "n5" were enriched in motifs where most of the swimming activity occurred within the 2 last seconds of the 5-second time window (S7D Fig).

For the 1-second time window behavioral motifs belonging to clusters "A1" to "D1," "F1" and "K1" are the most frequently identified across the wild-type data (Figs 3F and S7E and S16 Table). This is consistent with our empirical observations that *Ciona* larvae spend a considerable time slowly swimming or staying idle. We then asked how the different drugs we applied have affected the motif cluster distribution. Raclopride treatment results in a statistically significant reduction in the representation of cluster "A1" and quinpirole of cluster "D1" (Fig 3F and 3G and S16, S17, S41–S43, and S54 Tables) suggesting that dopamine signaling is important for moderately active swimming and dwelling behaviors. Clusters "E1" and "H1" that represent moderately active and active larvae respectively, show a similar profile in response to imipramine and methiothepin used in this study (Figs 3F, 3G, S7E, and S7F, and S16, S17, S41–S43, and S54 Tables). Our findings could be explained in multiple ways. The more likely possibility is that the 2 clusters contain similar motifs or that they are parts of a larger motif thus they often occur together. Another possibility is that they are generated by a common underlying cellular and/or molecular mechanism. In sharp contrast, the remaining inactive clusters "F1," "I1," "J1," and "O1" have little in common in their response across the drug treatments(Figs 3F, 3G, S7E, and S7F and S16, S17, S41–S43 and S54 Tables). Among the active clusters "L1," "M1," and "N1" are strongly modulated in the sense that several drugs show

statistically significant changes in the usage of these clusters relative to wild type (S41–S43 and S54 Tables). Raclopride is the only drug that results in a statistically significant up-regulation of cluster "M1" indicating that normally dopamine signaling serves to suppress sharp CW or CCW turns (S41–S43 and S54 Tables). Interestingly, fluoxetine shows very strong phenotypes across most of the 1-second active clusters where it significantly up-regulates cluster "L1" (Figs 3F, 3G, S7E, and S7F and S41–S43 and S54 Tables). This suggests that serotonin signaling likely suppresses startle-like behaviors and unidirectional tail flicking.

In the 5-second time window, we observed a uniform representation of most clusters in wild-type data, except for the active cluster "j5" (Figs 3H and S7F and S18 Table). What can be readily appreciated is that most of the motif clusters in the 5-second time window are more strongly regulated by the drug treatments in comparison to the 1-second time window. Raclopride treatment significantly increased the usage of the moderate activity cluster "f5" (S44–S46 and S55 Tables). Mianserin, on the other hand, showed a significant redistribution of cluster usage between the high activity cluster "j5" and the moderate activity "k5" that is enriched in motifs that show a late onset of swimming maneuvers (S44–S46 and S55 Tables).

### *Ciona* motor behavior can be modeled in terms of states and transitions

Having demonstrated that motif identification using matrix profiling is a potent method for identifying the basic behavioral building blocks of *Ciona* behavior, we next sought to expand our work by performing a systematic analysis of the organization of the behaviors that can be performed by the *Ciona* larvae. With an underlying assumption that behavior is modular and can occur across multiple timescales, we modeled our behavioral data with a simple Gaussian hidden Markov model (G-HMM), which is a state-based statistical model [48]. HMM provides a dynamical framework to identify the distinct behavioral modules that repeat over time at different timescales [8,49–52]. We implemented a 10-state HMM to model larval swimming across different experimental conditions (see Methods). We analyzed the means and standard deviations of the features for each of the states (Fig 4A) to show that the states β, γ, η, and κ have one or more input features with a distinctively higher standard deviation. This indicates that the larvae, while in any of these states, take a range of eigencoefficient features [41] and/or quirkiness, suggesting variation in postures that can be inferred as a result of active swimming. We then produced animations of skeleton movements in the arena for each of the states (S8 Fig). The animations agreed with our inference that states β, γ, η, and κ correspond to active swimming in the arena.

We also assessed if our model can predict underlying states across experiments and datasets. For this purpose, we labeled time series of input features by the inferred state obtained from the model prediction (Fig 4B). The model suggests that the larvae exhibit intermittent locomotion with bursts of swimming and substantial periods of dwelling. Using the inferred HMM states, we visualized the trajectories of the neck point along the arena for different experiments (Figs 4C and S9S–S9V). Importantly, these visualizations highlight a new behavior that resembles the beat-and-glide behavior observed in zebrafish larvae [53]. This is an intermittent form of swimming defined by tail beating followed by gliding during which the tail remains relatively motionless either straight or with a small amount of curvature. State "η," which is one of the actively swimming states, is the dominant "beating" state that is followed by one of the multiple gliding states "α," "δ," "ζ," "ι." This set of skeletons shows that the range of postures within each of the states is highly consistent. For example, states "α," "δ," "ε," "ζ," "θ," and "ι" represent different idle phases each with unique skeleton shapes. For states "α" and "ε," the sampled skeletons are close to a straight line, which reaffirms the observation in Fig 4A, where they have the highest quirkiness values with negligible variance suggesting that

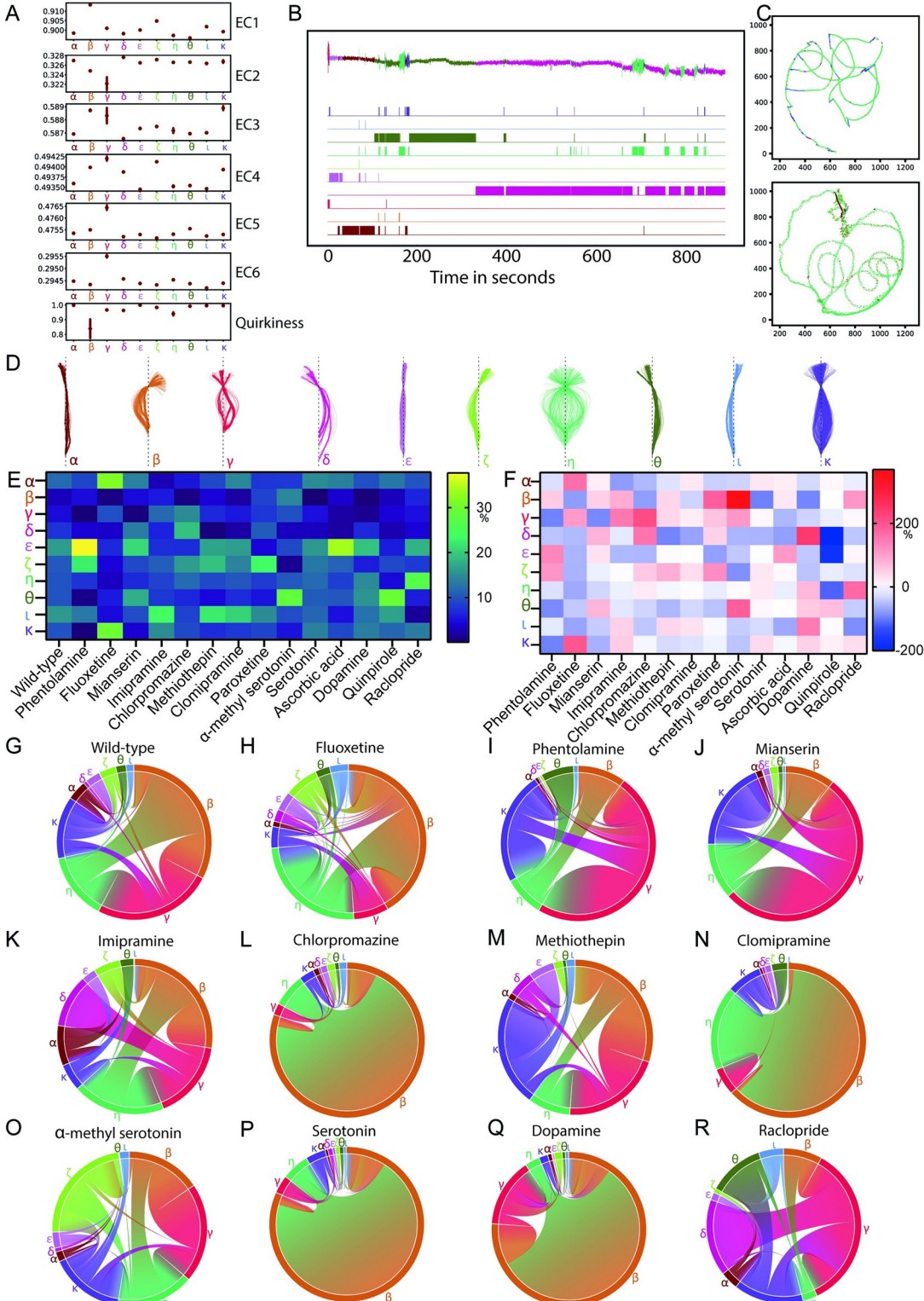

**Fig 4. HMM analysis reveals that behavioral transitions in *Ciona* are modulated by bioamines.** (A) Plots summarizing the observation probability distributions defined by the HMM model. For each of the states (horizontal axes), the mean of Gaussian distributions of each of the 7 input features is plotted along the vertical axes. The variance of the distributions is indicated by the error bars (underlying data can be found in S20–S22 Tables and https://doi.org/10.5281/zenodo.6761771). (B) HMM segments and clusters a time series into modules by identifying the underlying state for each time point in the series. On the top,

eigencoefficient EC1 (a time series) of a larva swimming is annotated with the different HMM states (uniquely color coded) identified in our analysis. (C) Two example tracks of the neck point of larvae in the arena colored according to the behavioral state identified by HMM. (D) Postures/skeletons were randomly sampled 90 skeletons from the dataset for each of the 10 different HMM states, aligned such that the neck points coincide and are collinear with tail-ends on a vertical line. (E) Heatmap visualization of the effects of drug treatments on the HMM-derived behavioral states (values in % can be found in S23 Table). (F) Heatmap visualization of percentage fold changes relative to wild type for the data shown in panel E (values in % can be found in S24 Table). Dopamine values are compared relative to ascorbic acid and not wild type. Drugs that resulted in a statistically significant up-regulation or down-regulation of the usage of HMM states are listed in S56 Table. (G–R) Chord diagrams showing HMM-derived behavioral state transitions for wild type and drug-treated larvae. Chord diagrams are presented in such a way that transition to all other states with probabilities greater than 0.001 are shown. Underlying data can be obtained from https://doi.org/10.5281/zenodo.6761771. EC1, Eigenciona 1; EC2, Eigenciona 2; EC3, Eigenciona 3; EC4, Eigenciona 4; EC5, Eigenciona 5; EC6, Eigenciona 6; HMM, hidden Markov model.

these states do not correspond to active swimming. Similarly, another trend revealed by our analysis is that skeletons belonging to states "β," "ζ," and "ι" are curved toward the left side of the vertical axis in Fig 4D and have higher values of EC1 feature. The wide range of skeleton shapes in state "η" (Fig 4D) agrees with the fact that active swimming and exploration of the arena would require the larva to take different postures at various stages within a single cycle of swimming motion. This is also confirmed by lower mean values of quirkiness for state η (Fig 4A).

Using the results from HMM, we examined how the distribution of distinct behavioral modules varies across the different drug treatments by calculating the percentage of representation of each of the states for different drug datasets. For example, we found that in fluoxetine-treated animals, states "α" and "κ" are significantly overrepresented in comparison to the wild type (Fig 4E and 4F and S23, S24, S47–S49, and S56 Tables). The active state "γ" was significantly up-regulated in several pharmacological treatments that block serotonin signaling including the SSRIs paroxetine and imipramine. This suggests that serotonin is an important signaling molecule for regulating active behavioral states. Compared to wild type, raclopride had a significantly decreased the representation of state "α" indicating that dopamine exerts an opposite effect to serotonin in the regulation of this state (Fig 4E and 4F and S23, S24, S47–S49, and S56 Tables).

Similarly, the transition probabilities obtained from an HMM allows us to look at how different drugs affect the transitions from one behavioral state to another. This is visualized using chord diagrams (Figs 4G–4R and S9A–S9C) and Markov transition graphs (S9D–S9R Figs). We found that in wild-type animals, transitions between states "β," "γ," and "η" were the most prevalent, forming a core transition module. Beyond this core module, active state "κ" acts as a "transit hub" for most behavioral state sequences that occur at a lower frequency. Importantly, state transitions do not occur in an all-to-all fashion (Figs 4G and S9D). For example, state "δ" interacts exclusively with state "γ" (S9D Fig). Our wild-type data suggest that certain states can interface with multiple other states, while other states may be more exclusive in their interactions. We found that the number of state transitions and the plausible pairwise combinations are sensitive to pharmacological treatment. Our chord diagrams and transition graphs reveal that clomipramine and imipramine reduce the transitions between behavioral states (Figs 4K, 4N, S9K and S9L). Conversely, fluoxetine and phentolamine result in an increase of transitions among behavioral states (Figs 4H, 4I, S9E and S9J). The "β" ←→ "γ" "η" ←→ module is preserved across all pharmacological treatments, and the same holds true for a number of state transitions (e.g., "α"-"κ," "ε"-"κ," "η"-"θ"); however, other transitions show drug-dependent "plasticity" (e.g., "γ"-"δ," "β"-"ι," "α"-"ε"). For example, state "δ" is solely interacting with the active state "γ" in the wild-type dataset (S9E Fig). Some drug treatments such as imipramine strengthen the transition between these 2 states (Figs 4F and S9L). However, in α-

methylserotonin-treated animals, "δ" interacts with state "θ" in (S9I Fig) or "η" in the case of clomipramine treatment (S5N Fig). Notably, we found that "δ" is not limited to interacting with 1 state but it can interact with multiple transition partners as observed in a subset of drug treatments (S9F, S9G, S9J, S9K, S9N–S9P, S9R and S9S Figs). Additionally, inferences can be made by combining the information from the percentage representations of states and the chord diagrams. For example, state "α" does occur at a very high percentage in larvae treated with fluoxetine; however, it can be seen from the chord diagram that the transition from "α" state occurs very rarely in these larvae, which indicates that they tend to stay idle and locked into state "α" for a long time.

## Low-dimensional spatiotemporal embedding identifies stereotyped actions in swimming larvae

While we have made assumptions that behavior is organized with units of repeated motifs (matrix profiling) or modules with Markov transitions (HMM), we wanted to additionally adopt a complementary approach aiming to uncover new structures in our behavioral data. For this purpose, we employed an approach similar to Berman and colleagues [7] to reveal stereotyped behaviors exhibited by the *Ciona* larvae.

For this approach, we used as the input the 6 eigencoefficient features [41] from a subset of the wild-type data where it was sampled based on speed (Fig 5A). The sampling was performed to ensure that the actively swimming epoch is well represented. First, to encode the temporal information such that clustering is applicable, we created a 180 dimensional feature set by computing the wavelet transformation of 6 eigencoefficient features at 30 different frequencies or scales (Fig 5B). Wavelet transformation allows us to create for each time point a feature that has information about its surrounding time points built into it, thereby creating a feature set where temporal information is preserved.

Next, to obtain a lower dimensional behavioral space from this feature set, we used t-distributed stochastic embedding (tSNE), which provides an embedding in which local structure is retained unlike many other dimensionality reduction techniques (Fig 5C). This embedding, which we can think of as the larval behavioral space, was then clustered using the DBSCAN algorithm (Fig 5D). We identified 6 distinct clusters corresponding to different stereotyped behaviors (Fig 5D). The DBSCAN, our algorithm also learned an additional outlier class that corresponded to less than 0.001% of the data points where cluster assignment was ambiguous (shown in blue in Fig 5D).

Our results show that the clusters identified by this method are coherent across datasets. For example, cluster 1 represents video frames where the larva is actively swimming and exploring the arena. This can be seen in the trajectories of the neck point across 2 different experiments where cluster 1 (orange) dominates the phases when there is movement across the arena (Fig 5E). On the contrary, cluster 2 (green) represents phases where the larvae were gliding in the arena (Fig 5E). By inspecting a number of animal trajectories, we were able to identify instances of beat-and-glide behavior like those observed using the HMM method. We provide further confirmation that the clusters we identified are coherent by randomly sampling skeletons across experiments for different clusters and plotting them such that the neck point and the end point are aligned on a vertical axis (Fig 5F). Here, cluster 2 has the least variations in the skeleton postures it represents. In addition, examining the trajectories and snippets of animations of skeleton movements uncovered that cluster 3 (red) represents sharp turns associated with transitions from a gliding phase (for example, cluster 2) to an actively swimming behavior (for example, cluster 1) (S10 Fig). In addition, from a detailed inspection of trajectories of the skeleton in the arena, we have identified that clusters 5 (brown) and 6

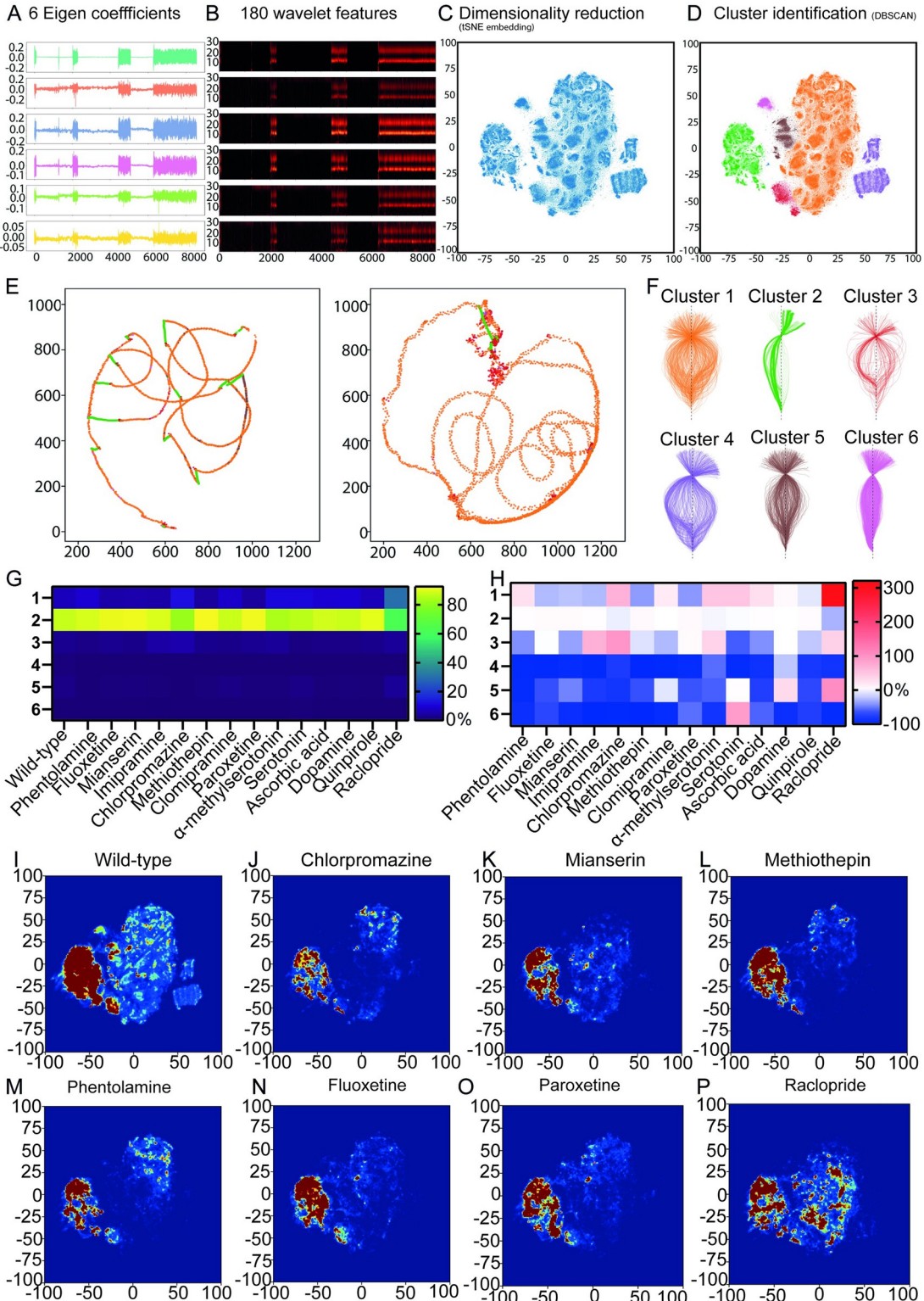

**Fig 5. Spatiotemporal embedding uncovers the influence of neuromodulators on stereotyped behaviors.** (A) We used the 6 eigencoefficients we previously obtained to define a 6-dimensional input feature space. (B) We computed wavelet transforms of these features over 30 frequencies for the wild-type dataset. Wavelet transform of the 6D eigencoefficient time series of panel A shown. (C) Dimensionality reduction was applied using tSNE to embed the 180-dimensional wavelet features into a 2D space of behavior. (D) We then used the DBSCAN algorithm to cluster into regions based on their density. (E) Examples of neck point

tracks for larval swimming labeled according to cluster identity. (F) Examples of skeletons aligned to the neck for each of the clusters. (G) Heatmap visualization of the effects of drug treatments on cluster usage (values in % can be found in S25 Table). (H) Heatmap representation of the percentage fold change relative to wild type for the data shown in panel (G) (values in % can be found in S26 Table). In panels G and H, dopamine values are compared relative to ascorbic acid. (I–P) 2D tSNE embedding of wild type and different drugs. The color is showing the density with which the different clusters are occupied (blue being lower and red higher). Underlying data can be obtained from https://doi.org/10.5281/zenodo.6761771.

(magenta) represent swimming with lower speeds compared to the higher speeds exhibited by clusters 1 (orange) and 4 (purple). A similar trend can be seen in Fig 5F where the former clusters have a narrower range of postures in comparison to a wider range of skeleton postures exhibited by the clusters 1 and 4.

Subsequently, we trained a kNN classifier on the clustering results and used this classifier to assign cluster membership to datapoints from the drug-treated dataset. This approach lets us compare how larvae across different experimental groups utilized the behavioral space (Fig 5G and S25 Table). In comparison to wild type, α-methyl serotonin-treated larvae utilized active swimming cluster 4 (purple) to a significantly lower extent (Figs 5G, 5H, 6I–6P, S10G, and S10H and S25, S50–S52, and S57 Tables). Cluster 1 usage was significantly up-regulated by raclopride and α-methyl serotonin (Figs 5G, 5H, 6J, 6P, S10J, and S6M and S20, S45–S47, and S52 Tables). In contrast, the antidepressant methiothepin showed a substantial reduction of cluster 1 (Figs 5G, 5H, 6K, 6L, 6N and 6O and S25, S50–S52, and S57 Tables). Sharp turns that occur when larvae transition from idle to swimming that correspond to cluster 3 are significantly up-regulated by imipramine and α-methyl serotonin (Figs 5G, 5H, and S10J and S20, S50–S52, and S57 Tables).

Finally, we mapped the transition probabilities between the behavioral clusters that allowed us to get insight into the organization of larval behavior. For example, we could infer that wild-type larvae executing the beat-and-glide behavior have multiple intermediate cluster options to transition from a "beating" phase (dominated by cluster1) to a "gliding" phase (cluster 2). However, exit from the gliding phase to the beating phase (in this case, primarily clusters 1 and 4) preferentially occurs through cluster 3, which is characterized by an asymmetrical swimming movement (S11A Fig and S3 Movie). The 2➜3 and 3➜4 transitions are "resistant" to almost all pharmacological perturbations, except for 3➜4 in chlorpromazine animals. However, the 3➜1 transition statistics are subject to modulation but several drugs affecting serotonin, noradrenaline, and dopamine signaling (S11F, S11H–S11K and S11N Fig). Additionally, pharmacological treatments can establish new interactions between clusters (e.g., 5➜4) (S11F and S11K Fig). These illustrate that *Ciona* locomotion has a certain degree of hierarchy and organization and that some of the transitions could be at least in part controlled by bioamine signaling.

## Light stimuli modulate postural dynamics and behavioral space occupancy

In addition to pharmacologically inhibiting bioamine neurotransmission, we sought to address how the presentation and removal of sensory cues such as light affect larval swimming behaviors.

Previous work has shown that ascidian larvae exhibit a shadow (i.e., looming-object escape) behavior as well as positive and negative phototaxis to white light [54–57]. Furthermore, it has been demonstrated that *Ciona* larvae can sense and respond to different wavelengths of light [58,59]. Interestingly, Nakagawa and colleagues have shown that the strength of the step-down (light off) response is dependent on the wavelength of light that was used [59]. Motivated by these studies, we examined how motor behavior changes when the larvae enter and exit a shadow stimulus period using white light as well as blue, green, and red light stimuli. Our

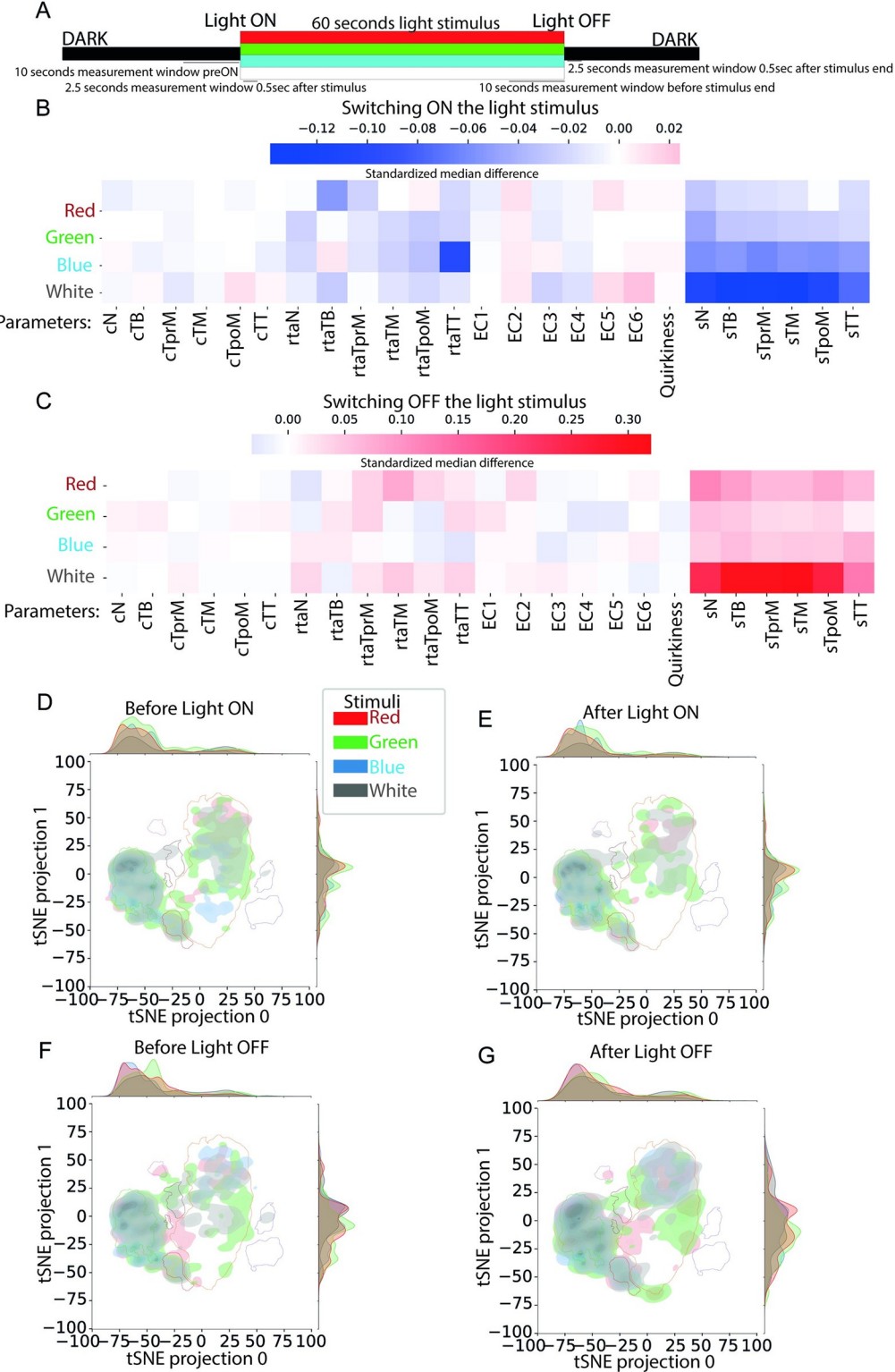

**Fig 6. Postural dynamics and behavioral space occupancy are modulated by different light stimuli.** (A) Design of the light stimulus: A 60-second long window of stimuli is presented at the 30th second of the experiment. Different types of light stimuli are used, namely red, green, blue, and white. We observe the changes at light ON event by comparing a 10-second window before the event (before light ON) and a 2.5-second long window after 0.5 seconds of the onset of light ON (after light ON). Similarly, we define 2 windows before and after the light OFF event to study the

changes in behavior when the animal exits the stimulus period. (Number of animals and frames used to generate this figure are indicated in S1 and S2 Tables.) The properties of the LEDs we used are shown in Table 2. (B) Effect of light ON event in the different biophysical features for each of the 4 stimulus types. The effect is measured in terms of the SMD in the feature values between the "before light ON" and "after light ON" intervals. (SMD values are shown in S27 Table.) (C) Effect of switching OFF the different light stimuli in terms of SMD is presented, similar to B (SMD values are shown in S28 Table). (D–G) 2D density plots showing the change in the pattern of occupancy of the 2D behavioral space across different stimuli before and after the light ON and OFF events. Boundaries of the 6 DBSCAN clusters are also shown on the plots (2D density plots for each stimulus individually are provided in S14 Fig). Underlying data can be obtained from https://doi.org/10.5281/zenodo.6761771. cN, curvature Neck; cTB, curvature Tail Base; cTM, curvature Tail Middle; cTpoM, curvature Tail post Middle; cTprM, curvature Tail pre Middle; cTT, curvature Tail Tip; EC1, Eigenciona 1; EC2, Eigenciona 2; EC3, Eigenciona 3; EC4, Eigenciona 4; EC5, Eigenciona 5; EC6, Eigenciona 6; rtaN, relative tan angle Neck; rtaTB, relative tan angle Tail Base; rtaTM, relative tan angle Tail Middle; rtaTpoM, relative tan angle Tail post Middle; rtaTprM, relative tan angle Tail pre Middle; rtaTT, relative tan angle Tail Tip; SMD; standardized median difference; sN, speed Neck; sTB, speed Tail Base; sTM, speed Tail Middle; sTpoM, speed Tail post Middle; sTprM, speed Tail pre Middle; sTT, speed Tail Tip.

SMD feature values reveal that once the larva enters the stimulus period, for a 2.5-second time window, the swimming speed decreases significantly when compared to a 10-second window defined before the stimulus period (Fig 6A and 6B and S27 Table). Removal of the stimulus gives rise to an opposite effect. A significant increase in speed is shown via the SMD speed values, in a 2.5-second window after the larva exits the stimulus period, when compared to a 10-second window during the presentation of stimulus period (Fig 6A and 6C and S28 Table). We show a similar trend for SMD relative tangent angle values (Fig 6B and 6C).

We additionally demonstrate a difference in responses across different light stimuli types: red, green, blue, and white. The differences in SMD speed values are significant for white light in both switching ON and switching OFF the stimulus (Figs 6B, 6C, S12, and S13 and S29–S36 Tables). Blue light has a significant effect while switching ON the light (Figs 6B and S12) but a smaller effect when switching OFF the stimulus (Figs 6C and S13). On the contrary, the effect of red light while switching OFF the stimulus is larger than during switching ON (Figs 6B, 6C, S12 and S13).

We then asked whether the presentation and removal of different color light stimuli influence the behavioral space explored by the larvae. As expected, the presentation of a light stimulus (ON) reduces the use of the active clusters in favor of the lower activity clusters 2 and 6 for white light stimulus (Figs 6D, 6E, S14D, and S14H and S32 Table). However, there seems to be light color-specific use of clusters. For example, in contrast to the white light ON period, during the red light ON period slow swimming behaviors under cluster 6 are not used. Our data suggests that the larvae gradually adapt to the continuous presence of the light stimulus especially in response to green and blue light as is evident from comparing the tSNE plots immediately after the onset (Figs 6E, S14E, and S14F) and prior to the end of the light stimuli (Figs 6F, S14I, and S14J). Subsequent removal of the light stimulus causes the larvae to increase their swimming activities and thus reuse the higher activity clusters of the behavioral space (Figs 6G and S14M–S14P).

## Discussion

### Motion tracking and comprehensive feature extraction at high-throughput, resolution, and reliability

In this work, we have characterized the feature-rich swimming behaviors of the protochordate *C. intestinalis* at an unprecedented level of detail. Previous behavioral studies from ours and other groups that investigated locomotion in *Ciona* [43,55,59–67], *Oikopleura* and amphioxus (reviewed in [20]), have been challenged by the lack of advanced tracking and analysis

methods, resulting in a limited quantitative characterization of the structural organization of the invertebrate chordate's motor behavior. For example, in our previous study (Rudolf and colleagues [43]) where we quantified larval behavior by estimating the position of its centroid (i.e., the center of mass), we were just able to obtain the centroid trajectory and speed of each larva in the arena. Such coarse tracking was sufficient to make meaningful comparisons of the swimming paths of the larvae and it allowed us to study how different rearing conditions (e.g., temperature) affect the basal behavioral repertoire of the animal. However, a major limitation of our previous study was that centroid tracking does not capture the larval body posture. Posture information, which includes the body midline, can be used to detect behaviors that might not be discernible from mere centroid data. In addition, postural features such as curvature can be used to reveal subtle phenotypes in biological relevant senses such as proprioception that would be impossible to quantify relying exclusively on centroid information. Thus, a major technological advancement of this work is the utilization of the lengthwise midline of the larvae obtained using the Tierpsy software [39], at high-throughput and reliability over the entire duration of the larval swimming videos. Through this approach, we measured biophysical features such as segment speeds, curvatures, and tangent angles, classically important parameters for describing motor behavior that were not available to us when we used centroid tracking [43]. In addition, we have derived a dimensionality reduced representation of *Ciona* body postures that we term "eigencionas." We show that just 6 basic shapes (eigencionas) can be combined in different proportions to reconstruct almost the entirety (approximately 97%) of *Ciona* postures during swimming. The use of eigenvectors as lower dimensional representations of posture has been well established in *C. elegans* [9,41,68,69], *Drosophila* [70,71], and zebrafish [72,73]. We believe that analogous to what has been done in these mainstream model organisms, eigencionas will be widely employed in future behavioral analyses in *Ciona* and other ascidians.

Behavior is a highly dynamic phenomenon that entails changes to an animal's posture over time. The realization that a major fraction of animal locomotion is low dimensional and stereotyped has sparked the development of multiple approaches to quantify stereotyped behavioral dynamics across several model organisms (reviewed in [3–5]). Our study is one of the few to the best of our knowledge to use multiple complementary approaches that impose a modular structure on the behavioral dynamics of *Ciona*.

## Motif analysis enables generation of a multiscale dynamic behavioral representation

Our first method searched for and extracted motifs from tracking data to generate a dynamic multiscale representation of *Ciona* motor behavior. Comprehensive inspection of the plethora of identified motifs has highlighted that existing manual approaches would most likely overlook motifs that represent subtle or apparently irregular yet repetitive behaviors. For example, in comparison to our findings in Rudolf and colleagues [43], our present study identifies some common behaviors (e.g. "twitching"); however, the current work has exposed the previously underappreciated wealth of CW and CCW exploratory swimming maneuvers, which may reflect the asymmetry of motor pathways in the *Ciona* larval connectome [21]. In addition, here we discover a novel startle-like maneuver, which may be associated with a pair of descending decussating neurons (ddNs) found in the motor ganglion. ddNs' ultrastructure, network connectivity, and synaptic connections have been elucidated in a recent study that has postulated that this neuron pair show network homology to vertebrate reticulospinal neurons and that their synaptic connectivity resembles that of the Mauthner cells [22], which underlie fish startle responses [74,75]. Indeed, we demonstrate in this work that serotonin

suppresses startle-like behavioral maneuvers. In line with this result, serotonin has been shown to regulate startle responses in zebrafish [76]. Our findings thus suggest that motor behavior as well as the underlying molecular and cellular players is conserved across invertebrate chordates and vertebrates.

## Model fitting preserves temporal component to expose motor modules and transitions

To further examine the modularity and transition structure in *Ciona* behavioral dynamics across diverse spatiotemporal scales, we have built an HMM. Using HMM, we have shown that *Ciona* larval locomotion can be decomposed into multiple distinct stereotyped locomotor states that can occur over a wide range of spatiotemporal scales. This framework provides the first probabilistic model (in terms of Gaussian distributions) for each of the distinct motor behavioral states of the *Ciona* larvae.

Despite a relatively simple nervous system equipped with a minimal number of neurons that make up the motor circuit [21,22], *Ciona* larvae exhibit multiple locomotor modes. In Rudolf and colleagues, we made a first attempt to generate a simple ontology of behavioral modes of swimming using agglomerative clustering of a minimal feature set based on centroid velocity vectors [43]. However, a confounding factor of the study was the lack of postural information, which would have enriched our dataset significantly. In addition, in contrast to our current study in Rudolf and colleagues, we were not able to explore the transition dynamics between different behavioral modes. The work presented here has identified different low activity states that are distinguishable by different resting postures adopted by the larvae and high activity states with distinctive swimming speeds and bending asymmetries. Interestingly, we found that most of the low activity states exhibit unilateral (either left- or right-handed) tail bending/flicking, while 1 active state shows unilateral tail bending. Sided flicking and swimming are likely generated by asymmetries in sensory input to the motor ganglion of the larva as suggested by the *Ciona* larval wiring diagram [21]. In addition, we have identified a new behavior that shows strong similarity with the beat-and-glide behavior that zebrafish larvae perform [53]. In zebrafish, there are suggestions that this behavior is at least in part by dopamine [77,78]. Our findings indicate that both serotonin and dopamine are important for executing this behavior. From an evolutionary perspective, it would be interesting to determine whether other invertebrate chordates such as amphioxus and *Oikopleura doica* are able to perform a beat-and-glide-like behavior and to which extent this behavior may be under the control of the same or different neuromodulators across chordates.

Another key feature of HMM is the ability to describe the organization of behavior across time in terms of transition probabilities. The transition structure in wild-type behavioral sequences revealed a core module composed of 2 asymmetric slow swimming states and an active symmetric swimming state that is dominant during exploratory behaviors. It also identified an active state ("κ") that acts as a hub for the less frequent transitions that occur between states outside this core module. Importantly, the time spent by a larva in each behavioral state and the transitions between states appear to be modulated by bioamines.

## Construction of a *Ciona* larval locomotor behavioral space

In parallel to employing the above methodologies, we have implemented a spatiotemporal mapping approach similarly to what has been used in mapping complex behavioral responses in *Drosophila* and mice [7,79,80]. Here, we present 6 annotated clusters of different behaviors that segregated to the different regions of the map.

Interestingly, as with the HMM method, the beat-and-glide behavior is also identifiable using the spatiotemporal mapping approach. Due to the smaller number of clusters generated by the latter method, it has been more straightforward to visually infer the cluster usage and the key transitions between clusters that are required to generate this behavioral maneuver.

In addition, using behavioral mapping, we could visualize behavior as a trajectory across a manifold and examination of the dynamics of the same. In this behavioral space, spontaneously swimming *Ciona* larvae can "navigate" between the 6 behavioral islands of stereotyped behaviors in defined manners. This approach has been particularly informative in the light stimuli experiments, where we demonstrate that the behavioral responses and adaptation observed in response to different light color stimuli do not result in global changes in the underlying spatiotemporal structure, but rather they arise from the selective use of modules and changes in the transition statistics. This suggests that the larval brain can alter the use of individual modules and the transition statistics to generate responses to novel situations (e.g., sudden presentation of a sensory stimulus). This is likely a conserved strategy among invertebrates and vertebrates used to produce complex behavioral actions in response to sensory cues [8,81].

## The proto-hypothalamic territory of *Ciona* may influence transitions between different behaviors using bioamines

As a result of the multiple analytical methods we have employed in our study, we have broken down relatively complex larval behaviors to simpler modules that can be assembled in different sequences to generate diverse behavioral output that is likely modulated by internal state changes during spontaneous swimming or in response to sensory cues, in our case different light stimuli. This strategy has been observed across vertebrates and invertebrates [6,82]. We demonstrate here that bioamines contribute to the modulation of the transition statistics and overall time spent in different forms of active exploratory swimming and locomotor periods defined by distinct forms of dwelling, gliding, and slow swimming. This is likely an evolutionarily conserved function [37,83]. Dopamine promotes dwelling, gliding, and slow swimming states, a phenomenon that has been observed in a number of organisms including zebrafish and xenopus [78,84]. On the other hand, serotonin and noradrenaline promote active exploratory swimming rich in CW and CCW turns. Notably, in mammals arousal and waking states are stimulated by serotonin and noradrenaline [85].

An obvious question that arises from our study is which cells and anatomical structures in the tadpole brain use bioamines to modulate the composition and organization of the larval behavioral repertoire? The dopaminergic cells are composed of a single-cell cluster called the coronet cells, which have been characterized molecularly and homologized to the vertebrate hypothalamus [60,86–89]. The same cells express the serotonin transporter (CiSERT) [60], though the rate limiting enzyme in serotonin synthesis tryptophan hydroxylase (TPH) is expressed in the vicinity of the motor ganglion and tail muscles [90]. Given that the vertebrate hypothalamus is also capable of modulating behavioral states by the secretion of neuromodulators [91,92], it is likely that *Ciona*'s proto-hypothalamic structure shares not only molecular but also functional similarities with its vertebrate counterpart.

## Comparison of complementary computational ethology methods used to quantify the behavioral repertoire of *Ciona*

Our first approach to this problem was to search in the data for highly repetitive fixed-length subsequences that the larvae employ to explore the arena. To this end, we searched for highly repeated fixed-length subsequences (motifs) in the dataset using matrix-profiling. Matrix-

profiling enables us to capture such motifs at predefined timescales in a computationally scalable and efficient manner. We employed it to find highly recurring behavioral motifs in 1-second and 5-second intervals across individuals and experimental conditions. Thus, the method allowed us to screen our large dataset for existence of motifs.

In our second approach, we used HMM so that we can find motifs across timescales, without being limited to pre-set intervals, which is an advantage over our matrix-profiling approach. Also, unlike matrix-profiling, which provides limited information about the non-motif regions, HMM allows to model and infer the underlying state of the animal at any point of time.

In our third approach, we examined if behavior could be modeled as a trajectory in a low-dimensional space, as suggested by Berman and colleagues [7]. This approach using wavelet transforms and t-SNE provided us the flexibility to sample the original input space such that different activity levels (defined by speed) are given uniform representation so that the low-dimensional space is not skewed by the low activity (dwelling) phase. Also, determination of the number of clusters (DBSCAN) is more verifiable since the behavioral space and the clusters can be visualized.

From an ethological point of view, all 3 approaches revealed the presence of a beat-and-glide-like behavior; however, only motif analysis uncovered that *Ciona* larvae exhibit a startle-like behavior. Gliding, active exploratory swimming, low speed swimming characterized by different extent of tail bending, and asymmetric active swimming are detected across all 3 methods; however, it is through motif analysis that we can best visualize the diversity of asymmetric maneuvers that can be performed by the larvae.

Drug treatments targeting bioamine signaling resulted in statistically significant changes in the representation of the clusters or states across all 3 approaches. The 1-second motif clusters and the HMM states were significantly up-regulated or down-regulated by a larger number of drug treatments compared to 5-second motif clusters and spatiotemporal embedding-derived clusters.

In sharp contrast to motif analysis, HMM and spatiotemporal embedding are suitable for revealing the stereotypy of the transitions that occur between different motor modules. While there is a discrepancy in the number of states (HMM = 10) and clusters (spatiotemporal embedding = 6), we still find some similar transition modules such as the "β" ←→ "γ" ←→ "η" (HMM) and the "2" ←→ "3" ←→ "1" (spatiotemporal embedding). Drug treatments altered the transition probabilities between different behavioral states or clusters, though a clear limitation in our approach is the fact that it is challenging to directly compare the effects of the drugs on transition probabilities across the 2 different methods (HMM and spatiotemporal embedding).

## Summary and outlook

This study shows that *Ciona* locomotor behavior is complex and flexibly structured, especially when we consider that the larval nervous system is equipped with less than 250 neurons. This complexity in behavioral output is likely conserved across tunicate larvae as indicated by earlier findings from 2 different tunicate clades the Aplousobranchia [56,93] and Appendicularia [20].

Our findings on the role of dopamine signaling in locomotion corroborate our earlier observations from Rudolf and colleagues where we showed that dopamine signaling is responsible for promoting low behavioral activity and reducing swimming speed [43]. Due to the higher-throughput and resolution of this study, we were able to provide additional insight on the role dopamine in regulating larval behaviors and to extend our study to serotonin and

noradrenaline signaling. We have discovered that these bioamines play a major role in the observed complexity and flexibility of the locomotor repertoire by modulating postural features, behavioral modules, and their transitions, during spontaneous swimming and in response to sensory stimulation. This is in line with studies across invertebrate and vertebrate species, suggesting that bioamines have an evolutionarily conserved functional role in modulating locomotor behaviors [30,35–37,52,83,94–96]. Future studies, combining our behavioral analysis pipeline, functional imaging, and genetic mutants for key genes involve in bioamine signaling will enable us to obtain a systems level understanding for the role of bioamines in modulating neural activity and behavior in *Ciona*.

While modern neuroscience has strongly benefited from the classic model systems, recent technological developments have encouraged the expansion of functional studies to nontraditional models [18,97].

We have now established a framework for a higher-throughput yet higher-resolution dissection of the behavioral repertoire of *Ciona*. Our experiments reveal that the analytical approaches we have taken are capable of systematically capturing known and new behaviors that were unidentified previously. The high sensitivity of our approach can be leveraged for extracting subtle phenotypes and mapping the contribution of individual neurons and molecules to behavioral structure through chemogenetics and genome editing. Ultimately, *Ciona* may serve as a key organism to identify evolutionary constraints and flexibility at multiple levels of behavioral organization and reveal fundamental principles of how molecules, neurons, and circuits generate the chordate behavioral repertoire.

## Methods

### Animal collection and rearing conditions

Gravid adult *C. intestinalis* were collected from the following site in Bergen: Døsjevika, Bildøy Marina AS, postcode 5353, Norway. The GPS coordinates of the site are as follows: 60.344330, 5.110812. Animals were kept in a purpose-built facility at 10˚C with a pH of 8.2 under constant illumination. Fertilization and embryonic development conditions were as previously described [43]. Age distribution of the larvae we assayed in terms of hours post hatching is indicated in S15 Fig.

### Egg fertilization, embryo, and larval rearing conditions

Egg collection, fertilization, and rearing were done following standard methods [98] with the exception of the rearing temperature that was set to 14˚C. Briefly, at least 2 healthy and gravid animals were used to extract sperm and chorionated eggs. Activated sperm was mixed with eggs, and these were kept together for 10 to 15 minutes. Once fertilized, the eggs were washed multiple times and split into three 9-cm petri dishes (SARSTEDT 82.1473) coated with agarose (Invitrogen, Ultra-Pure Agarose 16500–500). These plates were placed in a 14˚C incubator, and development of the embryos was monitored regularly until the onset of hatching, which occurred approximately 36 hours post fertilization. Throughout the experimental day, hatched larvae were kept at 14˚C. The average size (length) of the larvae used in our experiments was 115.10 pixels or equivalently 1,330.61 μm. We obtained larval length measurements from randomly selected skeletons across multiple videos, and we confirmed the measurements using ImageJ.

### Experimental setup

Each Ciona Tracker 2.0 is built using a DMK 33UP1300 (Imaging Source) coupled to an MVL75M1 lens and 2 C-mount extensions CML10 and CML25 (Thorlabs). To print the

custom-made parts of the behavioral setups, we used a Weistek WT280A 3D printer. Using multiple 3D-printed PLA moulds, we made agarose arenas. In brief, we filled a 35-mm petri dish (SARSTEDT 82.1135.500) with 9 ml of 0.8% agarose in ASW. While the agarose was still warm, we placed the 3D-printed moulds into the agarose-filled petri dishes and waited until the agarose had settled. At that point, we removed the mould and cleaned any spill overs of agarose. The agarose arenas were then hydrated with ASW. The resulting circular arenas had a diameter of 10 mm and a depth of 3 mm. The approximate volume of the arena was 240 mm$^3$. Note that while the animals are not constrained in 2D, the depth of the well (3 mm) is limiting the third dimension (depth) available to the animal. New arenas were prepared every day. The arena was nested inside a PLA ring with infrared LEDs (IR, peak emission 850 nm). These LEDs provided dark-field illumination of the animals while preventing stimulation of their photoreceptors. The illumination ring and the arena were rested on an underlayer that also hosted a waterproof thermometer model DS18B20 (Maxim Integrated). Videos were recorded using the IR sensitive monochrome DMK33UP1300 camera. An Arduino-based circuit, interfacing with a GUI written in Python provided light stimuli, PID-temperature control, and captured video stream. The software controlling all functionalities of the setup is available on GitHub: https://github.com/ChatzigeorgiouGroup/immobilize. Further information including STL-files for 3D-printed components and schematics for the electronics can be found in our Github: https://github.com/ChatzigeorgiouGroup/imMobilize/tree/master/Hardware. Individual *Ciona* larvae were filmed using an array of 5 modular Ciona Tracker 2.0 systems. These 5 trackers were housed in a temperature-controlled incubator (SANYO, Medicool) that maintained a constant temperature of 14°C.

### Experimental procedure

The experimental procedure in this study is largely based on the methodology employed by Rudolf and colleagues [43]. Agarose arenas were prepared fresh every evening for the next day's experiments. This was primarily done so that once the agarose had solidified, and the mould was carefully removed the arenas could be firstly inspected for structural defects (these typically could be air bubbles trapped in the agarose, broken/collapsed arena edges), and then they were hydrated overnight at 14°C with ASW to minimize the chances of dried out arenas that would affect the quality of the recordings.

Wild-type control videos (i.e., animals in ASW) were collected every day. Each drug was assayed at least on 3 different experimental days. All drugs besides dopamine were dissolved in ASW so the equivalent control was larvae in ASW (defined as wild type in the text and figures). Dopamine rapidly oxidized in ASW. We found that the only way to prevent this process was to include ascorbic acid at a final concentration 28 μm (Table 1) in the ASW and dopamine solution. Thus, when we assayed dopamine, the control animals were incubated in ASW plus ascorbic acid. Animals that were assayed in ASW plus ascorbic acid are not included in the wild-type dataset and they were exclusively used in the comparisons with dopamine-treated larvae. Therefore, wild type refers to control animals that were assayed in ASW only exclusively.

Hatched, swimming larvae were initially transferred from the original 9-cm plates to a fresh 6-cm plate containing either ASW or the drug that was going to be tested on the day and then immediately transferred individually to their arena that also contained ASW or one of the drugs. To transfer larvae, we used disposable 15-cm glass pasteur pipettes (91704012, Duran Wheaton Kimble). Once the larvae were mounted on the tracking setups, the telescopic covers were extended to shield the animals from the ambient light of the room. Video recordings were started at this point. Every round of tracking lasted for 30 minutes, and it involved the

**Table 2. Properties of LEDs used in this study.**

| Emitting color | Model | Wavelength (nm) | Luminous intensity (med) | Voltage (V) |
|---|---|---|---|---|
| Red | 13CBAUP | 620–630 | 550–700 | 1.8–2.2 |
| Green | 13CGAUP | 515–530 | 1,100–1,400 | 3.0–3.2 |
| Blue | 10R1MUX | 465–475 | 200–400 | 3.0–3.4 |

White light stimuli were delivered using a HALOSTAR 10 W 12 V G4 halogen lamp with a nominal luminous flux of 130 lm. Its spectral power distribution can be found here: https://docs.rs-online.com/7d94/0900766b8128288b.pdf.

simultaneous acquisition of videos from 1 control larva and 4 larvae incubated with a drug. Each animal was assayed over a period of 30 minutes, and this period was split into 4 recordings similarly to our previous study [43]. We recorded an initial 15-minute acclimatization period movie followed by three 5-minute movies. The same larva was never used to record across different ages; thus, we recorded each larva for a maximum period of 30 minutes.

Basic acquisition parameters were setup prior to the start of the experimental day using the acquisition software. These included the frame rate (30 fps), the camera exposure time (0.00390625 second), Gamma (value = 1), and IR light intensity (level 40). Subsequently, we completed relevant metadata fields on the software including the drug treatment if any, the hatching time of the larvae, the crowd size (in this case set to 1), the number and lengths of videos we would like to acquire.

### Light experiments

Light stimuli (white, red, green, or blue) were given for 1 minute starting at the 30th second and ending at the 90th second of the first 5-minute movie. Table 2 details the properties of the LEDs used to deliver the stimuli.

### Tierpsy analysis

Videos of larvae recorded using our behavioral setup was then analyzed with the help of Tierpsy software package to extract positional data [38]. The software segments the larval pixels from the background of the arena and identifies the 2 contours of the larvae. The software then calculates 49 equally spaced coordinates on the 2 contours such that the first pair of coordinates represents the tip of the head and the 49th pair represents the tip of the larval tail. The software also calculates the width of the larvae as the distance between the corresponding points on the 2 contours and uses this to calculate the midline (henceforth referred to as a skeleton) described by 49 coordinates.

### Feature extraction

Following the Tierpsy analysis, we calculate a set of biophysical features with an aim to quantitatively describe the movement of the larvae in the arena. In order to quantify the amount by which parts of the larval body deviates from a straight line while swimming, we calculated the curvature at each of the 49 points on the skeleton. Assuming the skeleton to be a differentiable curve, the curvature was measured as the rate of change of the curve's tangent angle with respect to its arc length (as defined in [39]). To calculate the numerical derivatives, we used a Savitzky–Golay filter of window length 15 and polynomial order 2 using the implementation in the scipy, so that the skeleton is approximated by a smooth curve.

To visualize the correlation of curvature values along the length of the larval body, we calculated the covariance matrix. Curvature from a subset of 231 experiments or larvae (where

Tierpsy software successfully identified the larvae from all of the frames in the videos) with a total of 2,290,901 frames were used to obtain the 49 × 49 matrix (Fig 1). The smooth structure of the correlation matrix was indicative of a strong correlation, and hence, an existence of a lower dimensional feature space. We performed an eigen decomposition of the covariance matrix (PCA) to obtain the eigenvectors (principal components) and eigenvalues (explained of each of the components). The 6 eigenvectors are referred to as eigencionas in the paper. The eigencionas were sorted by the eigenvalues and the 6 top eigencionas that explained 97% of the variance were selected to provide a lower dimensional description of the curvature of the larvae. Having defined the 6 eigencionas as features, we can calculate eigencoefficients EC1, EC2,. . ., EC6 (principal components scores) at each time point (or for each frame) that describes the posture of the skeleton.

We also calculate quirkiness as a scalar valued feature in the range of 0 to 1 indicating the eccentricity of the larval body, as explained in Tierpsy [39]. Speed at each of the 49 points of the skeleton is also calculated across time as the distance by which the skeletal point moves in the arena between 2 adjacent frames in the video.

We defined 7 distinct body parts or segments on the larvae by grouping the 49 skeleton points. Initially, we identified a point in the range of 4 to 22 along the skeleton where the contour width decreases sharply (local minima of the derivative of contour width along the skeleton) and defined it as the neck point. The change in contour width is characteristic of the neck, where the wide head region ends and the narrower tail of the larvae starts. The neck segment is defined such that it comprises 3 skeleton points with the neck point as the center. The points on the skeleton that lie anterior to this segment are hence grouped into a head segment, and the coordinates that lie after are grouped into a tail segment. The skeleton points tail segment is further divided into 5 segments, namely tail_base (TB), tail_pre_mid (TprM), tail_mid (TM), tail_post_mid (TpoM), and tail_tip (TT), such that a summary of movement of the tail can be obtained without limiting the degrees of freedom.

Following the definition of the 7 body segments, we calculated another postural feature, namely relative tangent angles. Initially, tangent angles were defined for each of the segments as the angle made by the line segments joining the end points of the segments (on the skeleton) with the x-axis of the video frame. We used the arctan2 function in the numpy package to compute the tangent angles (in radians) from the xy coordinates of the 2 end points of each of the segments. To obtain a measurement in the larvae's coordinate system rather than the global coordinate system of the arena, we computed the difference of these tangent angles with respect to the tangent angle of the head segment. These differences were then defined as the 6 relative tangent angles describing the posture of the skeleton, one for each of the segments from neck to tail_tip.

### Head rigidity determination

The larval swimming as seen from the collected videos suggested that the head segment exhibited some rigidity. We verified this observation by measuring the deviations of the skeleton points in the head segment from a straight line and comparing it with the tail region. For this, we measured the perpendicular distances of each of the skeleton points in the head region from a straight line joining the head tip and the neck point. For the comparison, we measured the perpendicular distances of each of the skeleton points in the tail region from a straight line joining the neck point with the tail tip point. The distribution of these distances measured across the videos in our wild-type dataset was then plotted for each of the skeleton points. We also verified from the curvatures at each of the 49 skeleton points across multiple videos has a smaller range in the head segment when compared to the tail region.

## Statistics

For each of the parameters/features and each of the experimental conditions, we tested the data for normality using the Shapiro–Wilk test with an alpha value of 0.05 (S12 and S33 Tables). Since the *p*-values were less than the alpha value, the null hypothesis that data is from a normal distribution was rejected. Hence, for comparison between different groups in the further analysis, we used nonparametric tests. In the case of comparison of features like curvature, speed, and relative tangent angles across the 6 body segments for the wild-type dataset, we used Wilcoxon signed-rank test with an alpha value of 0.05. We used this test under the assumption that feature values across segments (along the body) are dependent due to the anatomy of the larvae. Similarly, we used Wilcoxon for comparing the 6 eigencoefficient values. The *p*-values from the 2-tailed alternative were used to reject the null hypothesis that the median of the differences between 2 distributions is 0. In addition to the 2-tailed test, we also computed the *p*-values for 1-tailed (greater and less) tests for determining which group in a pair was significantly greater or lesser than the other (S3–S6 Tables).

For the comparison of features between each of the drugs against the wild-type group, we used Mann–Whitney U tests with a Bonferroni correction. The alpha value was set at 0.05/ 25 = 0.002 after Bonferroni correction. While the 2-tailed test is used to test the alternative hypothesis that 2 distributions are not equal, we used 1-tailed tests (greater and less) to test if 1 of the distributions is stochastically greater or less than the other. For Mann–Whitney U tests, we calculated the effect size by dividing the test statistic by the product of the number of samples in each of the 2 groups being compared. All the statistical tests were implemented in Python using the scipy package. The *N*, *p*-values, and test statistics values for each of the tests are provided in S2 and S13–S15 Tables. The calculated effect sizes for Mann–Whitney U tests have also been provided in S38–S40 Tables. Note that all drugs except dopamine were compared to wild type. Dopamine was compared to ascorbic acid, which was used as a solvent for dopamine to stabilize it and prevent oxidation.

For each of the methods, namely matrix profile with 1-second window, matrix profile with 5-second window, HMM and spatiotemporal mapping, we used Mann–Whitney U test to compare the difference in percentage use of a cluster or state for a drug with respect to control (wild type for all drugs except dopamine, ascorbic acid for dopamine). For each of the videos when a given drug is administrated, the percentage of frames where the larvae is identified to be in a particular state or cluster is calculated. This distribution of percentage usages is then compared to the corresponding distribution for its control group. A *p*-value of 0.05 is used to test significance. The results of the test for each of the 4 methods are provided in S41–S52 Tables.

## Matrix profile methods

Motifs were identified by calculating the matrix profile of the multidimensional time series of 7 curvature values along the skeleton of each animal, using the mstump algorithm implementation in the stumpy Python library. A rolling mean filter over 10 frames was applied to the time series prior to matrix profiling. Recurring motifs were defined as stretches of either 30 or 150 frames with the starting point at locations where peaks in the matrix profile are under a set threshold value of 8. This resulted in a set of 87,569 motifs over 30 frames and a set of 18,776 motifs over 150 frames.

The sets of motifs were subsequently clustered into 15 groups using the TimeSeriesKMeans clustering algorithm in the tslearn Python library. The cluster number was decided by adding clusters until the decrease in final model inertia started leveling out.

Clusters were annotated by generating gifs of the skeletons over the duration of the motifs for each cluster and manually confirm if there is an enrichment for a certain behavior within a cluster.

## HMM methods

We used a simple G-HMM to model our data. A Python-based open-source library hmmlearn was used to implement the model and the related algorithms. We used a feature set derived from 1,613 recording across multiple experimental conditions for training the model to capture a wide range of behaviors. The recordings were chosen such that Tierpsy software had successfully segmented the larvae for at least 80% of the frames. For the training the model, we chose to use the 6 eigencionas and the quirkiness features from the selected experiments. Thus, our training set consisted of a set of 1,613 sequences, each one of them being a 7-dimensional time series of varying lengths (durations).

The model was trained using the "fit" function of the hmmlearn library. The function essentially performs an expectation maximization (EM) algorithm to estimate the parameters of the HMM model from the time series data. The learned parameters include the state transition probabilities of the fitted model and the Gaussian distributions corresponding to each of the HMM states. Based on the learned model, the most probable state sequence for each of the 1,613 time series were obtained using the Viterbi algorithm implemented as the "predict" function in the hmmlearn library. This enabled us to visualize the underlying state at each time point (frame) for any given time series.

Different models were trained with the number of hidden states chosen as 6, 8, 10, 12, and 15. Also, 2 types of covariance matrices: "full" (or unrestricted) and "diagonal" were tested. On a qualitative inspection, it was observed that when the number of states was chosen as 6 or 8, the active swimming behaviors were not well distinguished. On the other hand, a choice of 12 or 15 states gave rise to learning of states that appear to occur in the dataset with a frequency of less than 1% of the frames. A 10-state model with full covariance was observed to model distinct active states without over segmenting the data and enabled us to model any dependencies between eigencoefficient features and quirkiness.

The means and variances from the Gaussian distributions of each of the 10 states of the fitted model were analyzed to characterize and distinguish each of the states. Following the inference of states for each of the frames in the dataset, the percentage of occurrence of each of the 10 HMM states (behavioral states) were calculated for each of the different experimental conditions (wild type, drugs, light stimuli, etc). We also recalculated the probability of state transitions for each of these experimental conditions separately.

In addition, we redid the HMM fitting twice by using the entire data as in our original model to verify if similar structure is learned each time a 10-state model is used. The mean and standard deviation of the Gaussian observation probabilities of the 3 models fit on the entire dataset are provided in supplemental figure S16A–S16C Fig. The corresponding transition probability matrices for the 3 models are visualized in the supplemental figure S16J Fig (original model), S16K, and S16L Fig (additional models). Similarly, we fit 3 models with 3 mutually exclusive datasets with 581 videos each, sampled agnostic to the treatment group. The results are provided in supplemental figure S16D–S16F and S16M–S16O Fig. We also fit 3 models with 3 mutually exclusive subsets of 580 videos with uniform distribution of wild type, drug treatment, and light stimuli cases, and the results are provided in supplemental figure S16G–S16I and S16P–S16R Fig. The underlying numerical data for S16 Fig can be found in S58–S81 Tables as well as https://doi.org/10.5281/zenodo.6761771.

## Spatiotemporal mapping methods

A total of 694 wild-type videos were selected for the analysis/finding a behavioral space such that Tierpsy software identified the larvae in at least 80% of the frames. (This dataset is a subset of the experiments selected for HMM.) We initially calculated Morlet wavelet transformations

of the 6 eigencoefficient feature vectors of these experiments, at 30 uniformly spaced frequencies in the range of 1 to 30 Hz, giving a set of 694 distinct time series each of 180 dimensions.

To prepare the input dataset, we sampled a total of 200,000 frames from 694 wild-type animals. The sampling was done such that the probability of being sampled is proportional to a weighted sum of speeds in the neck and 5 tail segments. The speed of the neck segment was assigned a 50% weightage, whereas the 5 tail segments had a 10% weightage. The sampling was used to ensure that the active swimming behavior is well represented in the input dataset. The 180 wavelet feature values corresponding to the sampled frames were then used as the input features to create a wavelet feature dataset of shape $200,000 \times 180$.

To obtain an interpretable visualization of the high-dimensional wavelet feature set, we initially calculated a 2D tSNE embedding (embedding1). TSNE embeddings and associated learning algorithms were implemented using the Python package openTSNE (https://opentsne.readthedocs.io/en/latest/). Embedding1 was initialized using a PCA-based initialization and cosine-based metric was used for distances. A 2-step optimization process was performed (using the optimize function in openTSNE) to learn the embedding1, where the first step was run with a perplexity parameter of 500, exaggeration value of 12 (early exaggeration phase) and was followed by a second phase with a reduced exaggeration value of 1.

To facilitate effective clustering and identify stereotyped behaviors from the dataset, we crafted a 4-dimensional vector using 3 distinct embeddings obtained using different parameter combinations with tSNE implementation in the Python-based openTSNE library. The 4-dimensional t-SNE space was created by combining the x and y dimensions of the embedding1, x dimension of embedding2 (perplexity = 250 and exaggeration = 3) and y dimension of embedding3 (perplexity = 750 and exaggeration = 2). This 4D space was clustered based on the density of datapoints (into 6 regions) using the Pythonic implementation of DBSCAN algorithm available in scikit-learn library. The algorithm also learned an outlier class and less than 0.001% of the data points were clustered as outliers. To assign any new or unseen data into one of the learned clusters in the behavior space, a kNN-based classifier was trained with the results of the DBSCAN algorithm. We used the KNeighborsClassifier implementation in scikit learn library with n_neighbours = 200 and distance-based weights. We then followed a similar pipeline of methods to obtain clustering results for all the frames from 1,613 experiments across experimental conditions using the precomputed embeddings. The clusters were then assigned to this processed dataset so that we obtain the behavioral cluster for all the frames in our dataset.

We then randomly sampled 100 skeletons for each of the clusters, translated and rotated them such that the coordinates of the neck point and the end (49th) point lie on a vertical axis (Fig 5B). Also, we visualized the trajectories of the neck points in the arena for a randomly selected set of experiments such that each point is labeled by the cluster into which it was classified. To analyze the differences in behavioral space occupied by the larvae under different experimental conditions, we computed 2D histogram smoothed by a Gaussian filter from the scatter plots of the behavioral spaces. These were plotted as 2D density graphs where the intensity is set to saturate at 0.8% of the maximum intensity (Fig 5D).

## Supporting information

**S1 Fig. Components of the Ciona Tracker 2.0 and process of generating an agarose arena.** (A) Multiple 35-mm petri dishes were placed on a cold block. (B) After pouring 0.8% agarose in ASW into the petri dishes, we clipped on the 3D printed PLA moulds and allowed the agarose to solidify. (C) Carefully removing the mould revealed a circular arena marked with an asterisk. (D) Multiple arenas were prepared daily and disposed of at the end of the experimental day. (E) View of a single Ciona Tracker 2.0 setup without a telescopic cover fitted. A cover

from a neighboring tracker is indicated by a white arrowhead. (F) Detailed view of the upper part of the tracker, with the camera secured with a 3D-printed camera holder (denoted with a C), the extension tubes CML10 and CML25 (indicated as ExT) (Thorlabs), the lens MVL75M1 (Thorlabs) (labeled as L). The white arrowhead points to the holder of the color LEDs and the IR filter. The black arrowhead points to the telescopic tube cover. (G) Close-up view of PLA ring with infrared LEDs (peak emission 850 nm). Black arrowhead indicates plastic arms that secure the PLA ring to the underlayer. White arrowhead indicates a mat black plastic that reduces unwanted reflections and provides a uniform black background. The gap in the PLA ring indicated by the asterisk is aligned with the agarose arena. (H) Close view of the 3D-printed box housing the electronics that control each tracker unit. (I) Close-up view of the color LEDs (white arrowhead) and white light (black arrowhead) arrangement. (J) Close-up view of an arena mounted on the PLA illumination ring. A water-soaked paper tissue in a plastic cap (white arrowhead) provides additional humidity during the recording preventing the arena from drying up. (K) 3D rendering of the PLA mould used to generate the agarose arenas. (L) 3D rendering of the individual parts needed to generate 2 PLA illumination rings housing the IR LEDs. (M) 3D rendering of the IR filter and the color LEDs. (N) 3D rendering of the plastic underlayer that is used to secure the PLA illumination ring and the thermometer. (O, P) 3D renderings of the electronics housing parts.
(EPS)

**S2 Fig. *Ciona* larvae have a rigid head.** (A) Schematic illustrating the measurement of deviations of skeleton points in the head and tail regions from a straight line. (B) Quantification of the perpendicular distance (measure of deviation) of skeleton points in the head and tail region from the corresponding straight line. (C) Examples from 3 different larvae while swimming indicating the range of curvature exhibited by each skeleton point across multiple frames. Underlying data can be downloaded from https://doi.org/10.5281/zenodo.6761771.
(EPS)

**S3 Fig. Comparison of curvature distributions across drug treatments.** Violin plots comparing the distribution of curvatures of 6 body segments for different drugs with wild type. Dopamine was compared to ascorbic acid. In this case, the control violin plot is colored green. Drugs with significant differences are shown. In all plots, drugs that showed significantly higher values of the feature are grouped together within a red-colored border, whereas drugs with significantly lower feature values with respect to wild type were grouped within a blue border. The data used to generate these plots are available in https://doi.org/10.5281/zenodo.6761771. We tested for normality using the Shapiro–Wilk test ($\alpha = 0.05$) (underlying values can be found in S12 Table). Wild-type and drug data features were compared using Mann–Whitney U tests with a Bonferroni correction ($\alpha = 0.002$) (please see S13–S15 Tables for the underlying numerical values). The relevant underlying data can be downloaded from https://doi.org/10.5281/zenodo.6761771.
(EPS)

**S4 Fig. Comparison of relative tangent angles distributions across drug treatments.** Violin plots comparing the distribution of relative tangent angles of 6 body segments for different drugs with wild type. Dopamine was compared to ascorbic acid. In this case, the control violin plot (i.e., left side) is colored green. Drugs with significant differences are shown. Note that in all the plots, drugs that showed significantly higher values of the feature are grouped together within a red-colored border, whereas drugs with significantly lower feature values with respect to wild type were grouped within a blue border. All relevant underlying data used to generate these plots can be accessed at https://doi.org/10.5281/zenodo.6761771. We tested for normality

using the Shapiro–Wilk test ($\alpha$ = 0.05) (see S12 Table for numerical values). Wild-type and drug data features were compared using Mann–Whitney U tests with a Bonferroni correction ($\alpha$ = 0.002) (see S13–S15 Tables for numerical values).
(EPS)

**S5 Fig. Comparison of speed distributions across drug treatments.** (A) Violin plots comparing the distribution of speed values exhibited in the presence of different drugs with the wild type. Dopamine was compared to ascorbic acid. In this case, the control violin plot (i.e., left side) is colored green. Note that in all the plots, drugs that showed significantly higher values of the feature are grouped together within a red-colored border, whereas drugs with significantly lower feature values with respect to wild type were grouped within a blue border. The data used to generate these plots are available in https://doi.org/10.5281/zenodo.6761771. We tested for normality using the Shapiro–Wilk test ($\alpha$ = 0.05) (S12 Table). Wild-type and drug data features were compared using Mann–Whitney U tests with a Bonferroni correction ($\alpha$ = 0.002) (S13–S15 Tables).
(EPS)

**S6 Fig. Comparison of eigencoefficients and quirkiness across drug treatments.** (A) Comparison of distribution of quirkiness values exhibited in the presence of different drugs with the wild type. (B) Violin plots showing the distribution of the 6 eigencoefficient features for different drugs in comparison with wild type. Dopamine was compared to ascorbic acid. In this case, the control violin plot (i.e., left side) is colored green. In all plots, drugs that showed significantly higher values of the feature are grouped together within a red-colored border, whereas drugs with significantly lower feature values with respect to wild type were grouped within a blue border. The data used to generate these plots are available in https://doi.org/10.5281/zenodo.6761771. We tested for normality using the Shapiro–Wilk test ($\alpha$ = 0.05) (S12 Table). Wild-type and drug data features were compared using Mann–Whitney U tests with a Bonferroni correction ($\alpha$ = 0.002) (S13–S15 Tables).
(EPS)

**S7 Fig. Visualization of skeletons, individual motifs, and drug effects for 1-second and 5-second motif clusters.** (A, B) Randomly selected skeletons of animals that correspond to each of the clusters shown in Fig 3D and 3E. (C, D) Individual motifs were grouped according to the motif cluster they correspond to. Panel C includes 1-second clusters, while panel D shows 5-second clusters. Motifs are color coded to show temporal progression (start➜end; violet➜red). (E–H) Raclopride shows much stronger effects on several clusters relative to the other drugs used in our screen. This means that in heatmaps Fig 3F–3I, it is hard to visually appreciate the differences between wild type and the other drugs. Thus, in panels E–H, we show the same heatmaps but we exclude Raclopride; (E) 1-second time window motif clusters representation (S16 Table). (F) Percentage fold changes relative to wild type for 1-second time window motif clusters (S17 Table); (G) 5-second time window motif clusters representation (S18 Table). (H) Percentage fold changes relative to wild type for 5-second time window motif clusters (S19 Table). The data used to generate these figures can be downloaded from https://doi.org/10.5281/zenodo.6761771.
(EPS)

**S8 Fig. Examples of skeleton trajectories corresponding to the 10 states identified by HMM.** (A–J) Shows 36 distinct samples of skeleton trajectories exhibiting each of the 10 behavioral states (identified by HMM). Each of the 36 motifs in the panels are of a minimum of 21 frames long. *Ciona* skeletons were sampled from our dataset that can be downloaded

**S9 Fig. Chord transitions illustrating behavioral state transitions inferred from HMM for a subset of drugs and example tracks colored according to HMM derived states.** (A–C) Chord diagrams showing behavioral state transitions inferred from HMM for (A) paroxetine, (B) ascorbic acid, (C) quinpirole. (D–R) Markov transition graphs for wild type and drugs. The graph represents the probability of transitions from 1 behavioral state to any other state as defined by the HMM model. Each of the nodes in the graph represents a behavioral state. Self-transitions are represented by arrows with matching colors. Any transition with a probability greater than 0.001 are shown with an arrow. Probability values are printed for all transitions that have a probability greater than 0.01. (S–V) Four example tracks of the neck point of larvae in the arena colored according to the behavioral state identified by HMM. The underlying data can be downloaded from https://doi.org/10.5281/zenodo.6761771.
(EPS)

**S10 Fig. Example skeleton trajectories for the 6 behavioral clusters inferred by spatiotemporal embedding and tSNE embedding of different drugs against neuromodulators.** (A–F) These panels show 36 distinct samples of skeleton trajectories exhibiting each of the 6 behavioral clusters inferred from the spatiotemporal embedding approach. Each of the 36 skeleton trajectories in the panels are of a minimum of 21 frames long. *Ciona* skeletons were sampled from our dataset that can be found in https://doi.org/10.5281/zenodo.6761771. (G–M) 2D tSNE embedding of different drugs. The color is showing the density with which the different clusters are occupied (blue being lower and red higher).
(EPS)

**S11 Fig. Transition graphs for wild type and neuromodulator drugs.** (A–N) Transition graphs for wild type and drugs. The graph shows the transitions from 1 behavioral cluster to any other cluster in terms of the probability of the transition as determined by our data. Each of the nodes in the graph represents a behavioral cluster. Self-transitions are represented by arrows with matching colors. All transitions with a probability greater than 0.001 are shown with an arrow. Probability values are printed for all transitions that have a probability greater than 0.01. The underlying numerical data can be downloaded from https://doi.org/10.5281/zenodo.6761771.
(EPS)

**S12 Fig. Comparison of different features in response to switching ON the light stimuli.** (A–D) Change induced in different feature values by switching ON the light stimulus: Comparison of distribution of curvature, relative tangent angles, speed, eigenciona coefficient, and quirkiness features before (tONbefore) and after (tONafter) the light ON event. Effect of 4 different stimuli—red, green, blue, and white shown from left to right. Note: In all the panels, significant results are highlighted with a higher opacity. S29 and S30 Tables provide the mean and standard deviation values for all plots shown in this figure. For statistical analysis, we used Shapiro–Wilk test for normality analysis. Subsequently, we performed Mann–Whitney U tests with Bonferonni correction; S33–S36 Tables provide statistical analysis for this figure. The underlying numerical data are available from https://doi.org/10.5281/zenodo.6761771.
(EPS)

**S13 Fig. Comparison of different features in response to switching OFF the light stimuli.** (A) Change induced in terms of curvature values by switching OFF light stimulus: Comparison of distribution of segment-wise curvature values before (tOFFbefore) and after (tOFFafter) the light OFF event. Effect of 4 different stimuli—red, green, blue, and white shown from left

to right. (B, C) Plots similar to A for relative tangent angles and speeds, respectively. (D) Effect of switching OFF the light stimuli in terms of change in 6 eigencoefficient features. (E) Change in quirkiness feature with the light OFF event. Note: In all the panels, significant results are highlighted with a higher opacity; S31 and S32 Tables provide the mean and standard deviation values for all plots shown in this figure. For statistical analysis, we used Shapiro–Wilk test for normality analysis. Subsequently, we performed Mann–Whitney U tests with Bonferonni correction. Statistical analysis for data included in this figure can be found in S33–S36 Tables. Underlying data can be extracted from https://doi.org/10.5281/zenodo.6761771.
(EPS)

**S14 Fig. Occupancy of behavioral space by each color stimulus shown individually.** (A–P) This figure shows each color stimulus is shown individually, the data is the same as in Fig 6. 2D density plots showing the change in the pattern of occupancy of the 2D behavioral space across different color light stimuli before and after the ON and OFF events. This figure shows each color stimulus individually, the data is the same as in Fig 6. Boundaries of the 6 DBSCAN clusters are also shown on the plots. Underlying data can be downloaded from https://doi.org/10.5281/zenodo.6761771.
(EPS)

**S15 Fig. Age distribution plots for animals used in this study.** (A–N) Histograms indicating the age distribution in hours post hatching (hph) for wild type and drug-treated larvae.
(EPS)

**S16 Fig. Alternative hidden Markov models.** (A) Means and standard deviations of the observation probability distributions for the original HMM model trained on the entire dataset (1,613 videos). (B, C) Means and standard deviations of the observation probability distributions for 2 additional 10-state HMMs trained on the same dataset as A. (D–F) Means and standard deviations of the observation probability distributions for the 3 models (10-state HMM) trained on the 3 mutually exclusive subset of the original dataset with 581 videos each. The 3 subsets have different distribution of wild type, drug treatment, and light stimuli cases. (G–I) Means and standard deviations of the observation probability distributions for the 3 models (10-state HMM) trained on the 3 mutually exclusive subset of the original dataset with 580 videos each, and the 3 subsets have uniform distribution of wild type, drug, and light cases. (J–L) The transition probability matrices corresponding to panels A–C. (M–O) The transition probability matrices corresponding to the models referred to in D–F. (P–R) The transition probability matrices corresponding to the models referred to in G–I. Underlying data can be found in S58–S81 and https://doi.org/10.5281/zenodo.6761771.
(EPS)

**S17 Fig. Examples of skeleton trajectories corresponding to the 10 states identified by the alternative HMM trials.** Postures/skeletons were randomly sampled from the training dataset for each of the 10 different HMM states, aligned such that the neck points coincide and are collinear with tail-ends on a vertical line for each of the additional HMM models trained. (A, B) Skeletons sampled for the 10 states as learned by the 2 additional models trained on the entire dataset. (C–E) Skeletons sampled for the 10 states as learned by the 3 models trained on the 3 randomly split subsets. (F–H) Skeletons sampled for the 10 states as learned by the 3 models trained on the 3 uniformly split subsets. *Ciona* skeletons were randomly sampled from our dataset that can be found in https://doi.org/10.5281/zenodo.6761771.
(TIF)

**S1 Movie. Example 1 of startle-like behavior exhibited by a *C. intestinalis* larva.**
(AVI)

**S2 Movie. Example 2 of startle-like behavior exhibited by a *C. intestinalis* larva.**
(AVI)

**S3 Movie. Animation of an animal exploring the tSNE behavioral space.**
(AVI)

**S1 Table. Number of animals used in our analysis.**
(XLSX)

**S2 Table. Number of video frames per condition assayed that we used in our analysis.**
(XLSX)

**S3 Table. P-values for data corresponding to Fig 1G.**
(XLSX)

**S4 Table. P-values for data corresponding to Fig 1H.**
(XLSX)

**S5 Table. P-values for data corresponding to Fig 1I.**
(XLSX)

**S6 Table. P-values for data corresponding to Fig 1P.**
(XLSX)

**S7 Table. (Separate file) Fig 2A SMD values for 25 features, calculated for drug-treated larvae relative to wild-type larvae.**
(XLSX)

**S8 Table. (Separate file) Fig 2B EC mean values for drug-treated and wild-type larvae.**
(XLSX)

**S9 Table. (Separate file) Fig 2B EC standard deviation for drug-treated and wild-type larvae.**
(XLSX)

**S10 Table. (Separate file) Mean values of violin plots for different drugs and features shown in S3–S6 Figs.**
(XLSX)

**S11 Table. (Separate file) Standard deviation values of violin plots for different drugs and features shown in S3–S6 Figs.**
(XLSX)

**S12 Table. (Separate file) Shapiro–Wilk tests for normality analysis of the data shown in S3–S6 Figs.**
(XLSX)

**S13 Table. (Separate file) Mann–Whitney U 2-sided tests with Bonferroni correction for statistical significance analysis of the data shown in S3–S6 Figs.**
(XLSX)

**S14 Table. (Separate file) Mann–Whitney U 1-sided (less) tests with Bonferroni correction for statistical significance analysis of the data shown in S3–S6 Figs.**
(XLSX)

**S15 Table. (Separate file) Mann–Whitney U 1-sided (greater) tests with Bonferroni correction for statistical significance analysis of the data shown in S3–S6 Figs.**
(XLSX)

**S16 Table. (Separate file) Quantification of 1-second time window motif clusters representation (shown as %) in wild type and drug datasets corresponding to heatmaps shown in Figs 3F and S7E (in this case, raclopride is not included in the heatmap).**
(XLSX)

**S17 Table. (Separate file) Quantification of percentage fold increase of 1-second time window motif clusters representation of drugs relative to wild type corresponding to heatmap shown in Figs 3G and S7G (raclopride is omitted in this heatmap).** Dopamine is compared to ascorbic acid in which it was dissolved.
(XLSX)

**S18 Table. (Separate file) Quantification of 5-second time window motif clusters representation (shown as %) in wild type and drug datasets corresponding to heatmap shown in Figs 3H and S7F (with raclopride omitted in this heatmap).**
(XLSX)

**S19 Table. (Separate file) Quantification of percentage fold increase of 5-second time window motif clusters representation of drugs relative to wild type corresponding to the heatmap shown in Figs 3I and S7H (where raclopride is left out).** Dopamine is compared to ascorbic acid in which it was dissolved.
(XLSX)

**S20 Table. (Separate file) Mean values of Gaussian distributions of each of the 7 input features corresponding to Fig 4A.**
(XLSX)

**S21 Table. (Separate file) The variance of the observed probability distributions of each of the 7 input features corresponding to Fig 4A.**
(XLSX)

**S22 Table. Probability values matrix for all possible transitions between behavioral states and self-transitions corresponding to Figs 4A and S16J.**
(XLSX)

**S23 Table. (Separate file) Percentage of representation of the HMM states for different drug datasets, corresponding to panels Fig 4E.**
(XLSX)

**S24 Table. (Separate file) Percentage fold change of representation of the HMM states for different drug datasets, corresponding to panels Fig 4F.**
(XLSX)

**S25 Table. (Separate file) Percentage of representation of the tSNE clusters for different drug datasets, corresponding to heatmap Fig 5G.**
(XLSX)

**S26 Table. (Separate file) Percentage fold change of representation of the tSNE clusters for different drug datasets, corresponding to panel Fig 5H.**
(XLSX)

**S27 Table. (Separate file) SMD values corresponding to Fig 6B.**
(XLSX)

**S28 Table. (Separate file) SMD values corresponding to Fig 6C.**
(XLSX)

**S29 Table. (Separate file) Mean values of violin plots for different color light ON stimuli shown in S12 Fig.**
(XLSX)

**S30 Table. (Separate file) Standard deviation values of violin plots for different color light ON stimuli shown in S12 Fig.**
(XLSX)

**S31 Table. (Separate file) Mean values of violin plots for different color light OFF stimuli shown in S13 Fig.**
(XLSX)

**S32 Table. (Separate file) Standard deviation values of violin plots for different color light OFF stimuli shown in S13 Fig.**
(XLSX)

**S33 Table. (Separate file) Shapiro–Wilk tests for normality analysis of the data shown in S12 and S13 Figs.**
(XLSX)

**S34 Table. (Separate file) Mann–Whitney U 2-sided tests with Bonferroni correction for statistical significance analysis of the data shown in S12 and S13 Figs.**
(XLSX)

**S35 Table. (Separate file) Mann–Whitney U 1-sided (less) tests with Bonferroni correction for statistical significance analysis of the data shown in S12 and S13 Figs.**
(XLSX)

**S36 Table. (Separate file) Mann–Whitney U 1-sided (greater) tests with Bonferroni correction for statistical significance analysis of the data shown in S12 and S13 Figs.**
(XLSX)

**S37 Table. (Separate file) Percentage of representation of the tSNE clusters for different light color stimuli ON and OFF periods, corresponding to Figs 6D–6G and S14A–S14P.**
(XLSX)

**S38 Table. (Separate file) Effect size for Mann–Whitney U 2-sided, corresponding to S3–S6 Figs.**
(XLSX)

**S39 Table. (Separate file) Effect size less for Mann–Whitney U, corresponding to S3–S6 Figs.**
(XLSX)

**S40 Table. (Separate file) Effect size greater for Mann–Whitney U, corresponding to S3–S6 Figs.**
(XLSX)

**S41 Table. Mann–Whitney U 2-sided tests with Bonferroni correction for statistical significance analysis of the data shown in Fig 3F and 3G.**
(XLSX)

**S42 Table. Effect size for Mann–Whitney U 2-sided, corresponding to Fig 3F and 3G.**
(XLSX)

**S43 Table. Summary Table indicating which drugs are significantly different (value 1) and which are not significantly different relative to control (value 0) corresponding to Fig 3F and 3G.**
(XLSX)

**S44 Table. Mann–Whitney U 2-sided tests for statistical significance analysis of the data shown in Fig 3H and 3I.**
(XLSX)

**S45 Table. Effect size for Mann–Whitney U 2-sided, corresponding to Fig 3H and 3I.**
(XLSX)

**S46 Table. Summary Table indicating which drugs are significantly different (value 1) and which are not significantly different relative to control (value 0) corresponding to Fig 3H and 3I.**
(XLSX)

**S47 Table. Mann–Whitney U 2-sided tests for statistical significance analysis of the data shown in Fig 4E and 4F.**
(XLSX)

**S48 Table. Effect size for Mann–Whitney U 2-sided, corresponding to Fig 4E and 4F.**
(XLSX)

**S49 Table. Summary Table indicating which drugs are significantly different (value 1) and which are not significantly different relative to control (value 0) corresponding to Fig 4E and 4F.**
(XLSX)

**S50 Table. Mann–Whitney U 2-sided tests for statistical significance analysis of the data shown in Fig 5G and 5H.**
(XLSX)

**S51 Table. Effect size for Mann–Whitney U 2-sided, corresponding to Fig 5G and 5H.**
(XLSX)

**S52 Table. Summary Table indicating which drugs are significantly different (value 1) and which are not significantly different relative to control (value 0) corresponding to Fig 5G and 5H.**
(XLSX)

**S53 Table. Summary Table for Figs 2B and S6B listing the drugs that resulted in significantly higher or lower EC values relative to controls.** The corresponding *p*-values for the drugs listed in this table can be found in S14 and S15 Tables.
(XLSX)

**S54 Table. Summary Table for Figs 3F, 3G, S7E, and S7G listing the drugs that resulted in significantly higher or lower representation of 1-second motif clusters relative to controls.**

Corresponding data are in S41–S43 Tables.
(XLSX)

**S55 Table. Summary Table for Figs 3H, 3I, S7F, and 37H listing the drugs that resulted in significantly higher or lower representation of 5-second motif clusters relative to controls.** The corresponding data are in S44–S46 Tables.
(XLSX)

**S56 Table. Summary Table for Fig 4E and 4F listing the drugs that resulted in significantly higher or lower representation of HMM states relative to controls.** The corresponding data are in S47–S49 Tables.
(XLSX)

**S57 Table. Summary Table for Fig 5G and 5H listing the drugs that resulted in significantly higher or lower representation of spatiotemporal embedding clusters relative to controls.** The corresponding data are in S50–S51 Tables.
(XLSX)

**S58 Table. (Separate file) Mean values of observed probability distributions of each of the 7 input features for additional 10-state hidden Markov model 2 trained on the entire dataset (1,613 videos), corresponding to S16B Fig.**
(XLSX)

**S59 Table. (Separate file) Variance values of observed probability distributions of each of the 7 input features for additional 10-state hidden Markov model 2 trained on entire dataset (1,613 videos), corresponding to S16B Fig.**
(XLSX)

**S60 Table. (Separate file) Transition probability matrix values corresponding to S16K Fig.**
(XLSX)

**S61 Table. (Separate file) Mean values of observed probability distributions of each of the 7 input features for additional 10-state hidden Markov model 3 trained on the entire dataset (1,613 videos), corresponding to S16C Fig.**
(XLSX)

**S62 Table. (Separate file) Variance values of observed probability distributions of each of the 7 input features for additional 10-state hidden Markov model 3 trained on the entire dataset (1,613 videos), corresponding to S16C Fig.**
(XLSX)

**S63 Table. Transition probability matrix values corresponding to S16L Fig.**
(XLSX)

**S64 Table. (Separate file) Mean values of observed probability distributions of each of the 7 input features for additional 10-state hidden Markov model 1 trained on randomly shuffled data, corresponding to S16D Fig.**
(XLSX)

**S65 Table. (Separate file) Variance values of observed probability distributions of each of the 7 input features for additional 10-state hidden Markov model 1 trained on randomly shuffled data, corresponding to S16D Fig.**
(XLSX)

**S66 Table. Transition probability matrix values corresponding to S16M Fig.**
(XLSX)

**S67 Table. (Separate file) Mean values of observed probability distributions of each of the 7 input features for additional 10-state hidden Markov model 2 trained on randomly shuffled data, corresponding to S16E Fig.**
(XLSX)

**S68 Table. (Separate file) Variance values of observed probability distributions of each of the 7 input features for additional 10-state hidden Markov model 2 trained on randomly shuffled data, corresponding to S16E Fig.**
(XLSX)

**S69 Table. Transition probability matrix values corresponding to S16N Fig.**
(XLSX)

**S70 Table. (Separate file) Mean values of observed probability distributions of each of the 7 input features for additional 10-state hidden Markov model 3 trained on randomly shuffled data, corresponding to S16F Fig.**
(XLSX)

**S71 Table. Variance values of observed probability distributions of each of the 7 input features for additional 10-state hidden Markov model 3 trained on randomly shuffled data, corresponding to S16F Fig.**
(XLSX)

**S72 Table. Transition probability matrix values corresponding to S16O Fig.**
(XLSX)

**S73 Table. Mean values of observed probability distributions of each of the 7 input features for additional 10-state hidden Markov model 1 trained on uniformly shuffled data, corresponding to S16G Fig.**
(XLSX)

**S74 Table. Variance values of observed probability distributions of each of the 7 input features for additional 10-state hidden Markov model 1 trained on uniformly shuffled data, corresponding to S16G Fig.**
(XLSX)

**S75 Table. Transition probability matrix values corresponding to S16P Fig.**
(XLSX)

**S76 Table. Mean values of observed probability distributions of each of the 7 input features for additional 10-state hidden Markov model 2 trained on uniformly shuffled data, corresponding to S16H Fig.**
(XLSX)

**S77 Table. Variance values of observed probability distributions of each of the 7 input features for additional 10-state hidden Markov model 2 trained on uniformly shuffled data, corresponding to S16H Fig.**
(XLSX)

**S78 Table. Transition probability matrix values corresponding to S16Q Fig.**
(XLSX)

**S79 Table. Mean values of observed probability distributions of each of the 7 input features for additional 10-state hidden Markov model 3 trained on uniformly shuffled data, corresponding to S16I Fig.**
(XLSX)

**S80 Table. Variance values of observed probability distributions of each of the 7 input features for additional 10-state hidden Markov model 3 trained on uniformly shuffled data, corresponding to S16I Fig.**
(XLSX)

**S81 Table. Transition probability matrix values corresponding to S16R Fig.**
(XLSX)

## Acknowledgments

We thank Mie Wong for comments on the manuscript.

## Author Contributions

**Conceptualization:** Athira Athira, Daniel Dondorp, Jerneja Rudolf, Marios Chatzigeorgiou.

**Data curation:** Athira Athira, Daniel Dondorp, Jerneja Rudolf, Olivia Peytral, Marios Chatzigeorgiou.

**Formal analysis:** Athira Athira, Daniel Dondorp, Jerneja Rudolf, Marios Chatzigeorgiou.

**Funding acquisition:** Marios Chatzigeorgiou.

**Investigation:** Athira Athira, Daniel Dondorp, Jerneja Rudolf, Olivia Peytral.

**Methodology:** Athira Athira, Daniel Dondorp, Jerneja Rudolf, Olivia Peytral.

**Project administration:** Marios Chatzigeorgiou.

**Resources:** Marios Chatzigeorgiou.

**Software:** Athira Athira, Daniel Dondorp, Jerneja Rudolf.

**Supervision:** Jerneja Rudolf, Marios Chatzigeorgiou.

**Validation:** Athira Athira, Daniel Dondorp, Jerneja Rudolf, Olivia Peytral, Marios Chatzigeorgiou.

**Visualization:** Athira Athira, Daniel Dondorp, Jerneja Rudolf, Marios Chatzigeorgiou.

**Writing – original draft:** Athira Athira, Daniel Dondorp, Marios Chatzigeorgiou.

**Writing – review & editing:** Athira Athira, Daniel Dondorp, Jerneja Rudolf, Olivia Peytral, Marios Chatzigeorgiou.

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
