## [Editor Report · Decision Letter 0]

1 Feb 2022

Dear Dr Chatzigeorgiou, 

Thank you for submitting your manuscript entitled "Comprehensive analysis of locomotion dynamics in Ciona intestinalis reveals how neuromodulators flexibly shape the behavioral repertoire of a protochordate." for consideration as a Research Article by PLOS Biology.

Your manuscript has now been evaluated by the PLOS Biology editorial staff, as well as by an academic editor with relevant expertise, and I am writing to let you know that we would like to send your submission out for external peer review.

Once your full submission is complete, your paper will undergo a series of checks in preparation for peer review. Once your manuscript has passed the checks it will be sent out for review. To provide the metadata for your submission, please Login to Editorial Manager (https://www.editorialmanager.com/pbiology) within two working days, i.e. by Feb 03 2022 11:59PM.

If your manuscript has been previously reviewed at another journal, PLOS Biology is willing to work with those reviews in order to avoid re-starting the process. Submission of the previous reviews is entirely optional and our ability to use them effectively will depend on the willingness of the previous journal to confirm the content of the reports and share the reviewer identities. Please note that we reserve the right to invite additional reviewers if we consider that additional/independent reviewers are needed, although we aim to avoid this as far as possible. In our experience, working with previous reviews does save time. 

If you would like to send previous reviewer reports to us, please email me at ggasque@plos.org to let me know, including the name of the previous journal and the manuscript ID the study was given, as well as attaching a point-by-point response to reviewers that details how you have or plan to address the reviewers' concerns. 

Given the disruptions resulting from the ongoing COVID-19 pandemic, please expect some delays in the editorial process. We apologise in advance for any inconvenience caused and will do our best to minimize impact as far as possible.

Kind regards,

Gabriel

Gabriel Gasque

Senior Editor

PLOS Biology

ggasque@plos.org

---

## [Decision Letter · Decision Letter 1]

9 Mar 2022

Dear Dr Chatzigeorgiou,

Thank you for submitting your manuscript "Comprehensive analysis of locomotion dynamics in Ciona intestinalis reveals how neuromodulators flexibly shape the behavioral repertoire of a protochordate." for consideration as a Research Article at PLOS Biology. Your manuscript has been evaluated by the PLOS Biology editors, an Academic Editor with relevant expertise, and by three independent reviewers.

You'll see that all three reviewers are broadly positive about your study; however , they each raise a number of concerns which can probably be addressed largely by substantial re-writing and clarification of the paper, especially with the broad readership of PLOS Biology in mind.

In light of the reviews (below), we will not be able to accept the current version of the manuscript, but we would welcome re-submission of a much-revised version that takes into account the reviewers' comments. We cannot make any decision about publication until we have seen the revised manuscript and your response to the reviewers' comments. Your revised manuscript is also likely to be sent for further evaluation by the reviewers.

We expect to receive your revised manuscript within 3 months. 

**IMPORTANT - SUBMITTING YOUR REVISION**

*Re-submission Checklist*

*Published Peer Review*

*PLOS Data Policy*

*Blot and Gel Data Policy*

Sincerely,

Roli Roberts

Roland G Roberts PhD

Senior Editor

PLOS Biology

rroberts@plos.org

REVIEWERS' COMMENTS:

Reviewer #1:

I enjoyed reading and reviewing this paper. This project fills the need for more comparative, quantitative studies of neurobehavior beyond the most popular model organisms. The computational ethology is performed using up-to-date techniques and covers the behavioral space comprehensively. While the paper may read more like a PLOS Computational paper than a PLOS Biology paper, I think the broader readership of PLOS Biol. would appreciate this work. It may for many researchers be a model example for their own attempts to quantify and publish behavior outside of the mainstream model organisms.

Note that I am not an expert in C. interstinalis or very familiar with many of the pharmacological agents used in this study.

Comments:

1) I would like some more detail on the experiment. How were the animals prepped? I didn't see a reference to any "standard methods." Were recordings made immediately after preparing samples or were they left for some incubation time? How long did the sessions last? Were controls randomized across the data collection or did the wild-type (control) collection occur separately from the pharmacological experiments? In general when "control" is used I would like a clarification if this was a control for the specific type of experiment or was this just the wild-type data presented earlier. 

Re: the raw data and the imaging. I may have missed this, but what are the dimensions of the larvae and the dimensions of the container? The swimming is constrained to 2D I guess, but how? Is this due to a physical or behavioral barrier (e.g. liquid height)? How are the larvae illuminated for imaging (e.g. ring light?).

2) Fig.1. I don't think it is at all useful to show the GUI of the software used to capture and process images. It would be much more informative to get a very good schematic or image of the set up and a very good image of the raw data (i.e. larvae before processing).

3) The authors state that "given our assumption that the head segment is rigid." Can't this be shown rather than assumed?

4) Fig 3B. It's not clear what is being plotted here. Is it a specific skeleton segment? Center of mass?

5) re: light stimuli. As a reader not familiar with this organism, I wonder what the purpose was to use red, green, and blue light in addition to white light.

6) Since this PLOS Biol. and not PLOS computational biol. I wonder if there are certain descriptions that may be written in less technical language. I realize that this cannot be done for the entire text and so one might argue that the technical level is what it is and the reader will have to manage. Using terms like Principal Component Analysis in the text right beside the more technical description of "performing an eigenvalue decomposition of the covariance matrix" may give less advanced readers "a hook" to pursue a reference to help them with their understanding.

Minor comments:

In general I found the figures to be good, but there are many readability issues throughout. In many cases the axis labels are inconsistent in size and need to be enlarged. For example, Fig 1b has readable x-axis labels (EC1, EC2, etc) but the y-axis labels (-0.2, -0.1, etc) are very small even though there is plenty of space to increase the size. This happens throughout where the label sizes are mixed and often one axis label is very difficult to read.

The following are very minor typos or inconsistent references to chemicals. Instead of trying to map these out in the text, I think these can be found by the authors using the search function.

"Segments speeds'" should be "segments' speeds"

"Rapid holt" — "Rapid halt"?

Fig 3 F and Fig 3F (inconsistent figure referencing)

"wild type animal" - "wild-type animal"

Berman et al[7] - Berman et al.[7]

"eigen coefficient features" - "eigen-coefficient features"

"Mammping" - "mapping"

Chemicals are sometimes written as "ascorbic acid" and "Ascorbic Acid"

"python" maybe should be capitalized.

I scanned the supplemental information. It is not clear to me that all the movies are important to show beyond the plotted supplementary figure. 

Reviewer #2:

"Comprehensive analysis of locomotion dynamics in Ciona intestinalis reveals how neuromodulators flexibly shape the behavioural repertoire of a protochordate" by Athira et al. takes a series of computational ethology methods applied to other organisms and applies them to Ciona intestinalis larvae. This is worthwhile because, as the authors argue, Ciona has a compact nervous system and is a close relative of vertebrates, putting it at a unique niche with respect to evolution and neural complexity. At the same time, its behaviour has not yet been analysed using the latest computational tools. The quantification and visualisation of its behavioural repertoire and its modulation by light and drugs is therefore a useful contribution.

The paper is mostly clear, but I think it has two main weaknesses. The first is that three different methods are applied to the data but there is very little discussion of how the results relate to each other. Are there any advantages to one approach over another? Are they complementary or orthogonal (or something else) in their outputs?

The second major weakness of the work is that the results of the different quantification methods are difficult to interpret without statistical tests or some other measure of their informativeness. For example, there are many claims about a drug affecting some aspect of one of the quantification methods, but no indication of whether the result is due to a drug effect, noise, a batch effect, or something else. Specific cases are listed below.

Finally, an issue in multiple places, again listed below, is the presence of unsubstantiated claims that are not supported by the data. I believe these can be mostly addressed with re-writing, but together they may substantially change the paper, including the second half of the abstract.

Specific points:

-"which are likely left undetected with less sensitive methodologies". Without being specific or comparing to other methods, this claim should be removed.

-"Curvature is defined as inversely proportional to the radius of the osculating circle at a given point on the skeleton". The circles in the schematic figure seem to be oriented the wrong way, i.e. showing the opposite curvature to the skeleton.

-"software[38]" This software was first published in Javer et al. (2018) Nature Methods.

-"a scalar measure of eccentricity". Eccentricity is a scalar. Perhaps a related measure is better.

-When calculating the eigencionas in the results, it would make sense to cite the original eigenworms paper. I realise it is cited elsewhere already, but I would cite it here as well. The same goes for the results sections on motifs and the HMM.

-"Like EC1 and in contrast to EC2, EC3 exhibits a very strong positive trend across most drug treatments." Which of these changes are significant?

-"recurring motives and anomalies in the data" -> "recurring motifs and anomalies in the data"

-Figure is too big/labels too small. Even in the large tiff version of the figure, most detail is lost in the time series plots and the labels are difficult to read.

-"inertia of the model considerably less". I understand the idea but I think this could be rephrased to be more specific.

-"Raclopride treatment results in a strong reduction in the representation of Clusters "A1" to "D1"". for this and all of the related claims, which changes are significant? By eye, it looks like the effect of Raclopride might be the only significant one.

-" indicating that the behavioural motifs included in these clusters may be generated by a common underlying cellular and/or molecular mechanism." It's also consistent (and seems more likely) that they are simply similar motifs or are part of a larger motif so that they often occur together.

-"underlying states consistently across experiments and datasets." Rather than consistency across experiments, what follows seems to be a manual annotation. It might be more useful to quantify how similar the states are when the data are resampled or when different experiments are analysed independently.

-"we demonstrate that in Fluoxetine, states "α" and "κ" are overrepresented in comparison to the wild type". Are these differences significant?

-"Our results show that the clusters identified by this method are coherent across datasets. For example, cluster 1 consistently represents video frames where the larva is actively swimming and exploring the arena". As before, this doesn't really demonstrate consistency of the state. If one were to simply define a state as all cases where the larvae are slower than a threshold, then confirming that the larvae in that state are indeed slow doesn't provide evidence that the state is meaningful.

-"is more comprehensive than the discrete biophysical features such as segment speeds, curvatures and tangent angles; classically important parameters for describing motor behaviour." This may be true in some sense, but it needs to be made specific and supported by data. I would also drop "remarkably" from the start of the next sentence given the previous work cited in the sentence after that.

-"reveal its modular structure". It's more accurate to say that the methods impose a modular structure on the behaviour.

-"Model fitting preserves temporal component to expose biologically meaningful motor modules and transitions". Without confirming that the changes in module use are significant, it's difficult to argue they are biologically meaningful.

-"we have shown that Ciona larval locomotion can be decomposed into 10 distinct stereotyped locomotor states". I would rephrase this, since the statement would still be true for any number of states.

-"This richness of locomotor states comes as a surprise, since they are the output of a relatively simple nervous system equipped with a minimal number of neurons that make up the motor circuit of the larva[21, 22]". Again, given the previous applications of these methods to other systems, I would remove the reference to surprise. 

-"Our findings show that both serotonin and dopamine regulate the transition statistics between the beating and gliding states, pointing to a more complicated modulation of the beat-and-glide behaviour across species compared to what was previously thought." Possibly, but could be off-target effects of the compounds. Also, the null model that an important neurotransmitter has no effect on, e.g. beat-and-glide, seems unlikely to start with. More detail of what the 'previous expectations' are and whether these results contradict them would be required to support this statement.

-"states appear to be selectively modulated by bioamines." This statement is also not supported by the data. Selectively would imply that these changes are seen when levels of bioamines are changed but not other perturbations. In fact, given that the model output is basically time spent in states and the transition probabilities between them, anything that effects their behaviour would change one or both of these.

-"That, the key clusters and transitions involved in the behaviour were preserved under all pharmacological treatments suggests that these features are controlled by neuromodulators other than bioamines e.g. GABA and Glutamate. However, the transition statistics were demonstrated to be affected by dopamine and serotonin (which is in line with the HMM findings), indicating that bioamines likely perform the fine sculpting of this behaviour," It is also possible that these drugs do not accumulate in the larvae at a high enough concentration to act or that their targets are not sufficiently conserved to have the expected effect. The discussion of fine sculpting vs some other kind of change should be clarified or removed.

-"Here we have shown that Ciona larvae are able to use a finite number of behavioural modules". This isn't a result so much as a property of the analysis method.

-"here that bioamines act as behavioural switches" This should be made precise and supported by data or removed.

-"our findings indicate that these behavioural modules represent a class of druggable targets". I would rephrase this. Druggable targets would normally refer to a protein target not a behaviour.

Reviewer #3:

[identifies himself as Takehiro G. Kusakabe]

The authors' group has been using the Ciona intestinalis larva as a model animal for neurobiology and behavioral study. In this manuscript, the authors extended their previous studies to computational modeling of swimming dynamics and applied the method to pharmacological evaluation of bioamines effects and to analyze behavioral response to photic stimuli. The authors conducted high-throughput analysis of a large number of larvae and revealed that postural variance of Ciona larvae can be classified into 6 basic shapes, which they call "Engenciona". Then, the authors computationally analyzed temporal patterns of posture series, and compared the patterns between control larvae with experimental larvae (drug treated and different light stimuli). Their findings suggests roles of bioamines in specific neural and behavioral processes. The authors also propose possible conservation between Ciona larval behavior and vertebrate behavior. This study is highly original and provides a novel platform for studying neural and behavioral mechanisms of this emerging model organism. Therefore, this work potentially has a substantial impact on various biological fields, including neurobiology, physiology and ethology. However, there are some problems which should be solved before its acceptance for publication in PLoS Biology.

My major concerns are insufficient descriptions of experimental methods and unclear explanation of relationships between this study and a previous work by the same group (Rudolf et al. Sci. Rep. 2019).

1. The authors only provide "Drug concentrations Table" to explain their pharmacological assays. They should provide detail information for each chemicals, such as manufacturers, product codes, and solvents (DMSO, ethanol, etc.).

2. It is difficult to understand detailed design and use of their experimental setup (Ciona Tracker 2.0) for behavior recordings from a short sentence on the first page of Methods and relatively low resolution images presented in Fig. 1A. For example, what type of container did they use to keep the larva during recording? Petri dish or custom-made container? What size? Amount of medium (sea water?)? Any lid?? What temperature? Light source (infrared?) for visualization?

3. This group previously reported a paper of similar subject (Rudolf et al. Sci. Rep. 2019). The present manuscript refers to this previous paper (ref. 52), but it is unclear what was previously found or achieved and what are new in the present study. These points should be explained.

Minor points:

4. It is unclear the merit of naming "Eigenciona" for basic postural patterns. The authors use various inconsistent expressions, such as eigenvectors, eigen vectors, eigen features, Eigenciona vectors, Eigenciona, eigen Cionas, eigen-cionas, etc.

5. Last paragraph of Introduction, 7th line: lavae  larvae

6. In the paragraph after the legend for Fig. 2: two "for EC2" are contained in a sentence. It may be redundant.

---

## [Editor Report · Decision Letter 2]

22 Jun 2022

Dear Dr Chatzigeorgiou,

Thank you for your patience while we considered your revised manuscript "Comprehensive analysis of locomotion dynamics in Ciona intestinalis reveals how neuromodulators flexibly shape the behavioral repertoire of a protochordate." for publication as a Research Article at PLOS Biology. This revised version of your manuscript has been evaluated by the PLOS Biology editors and the Academic Editor.

Based on our Academic Editor's assessment of your revision, we would like to move towards publication of this manuscript. To do so, we will need you to address the following data and other policy-related requests listed below. We'd also ask that you consider a minor title change:

"Comprehensive analysis of locomotion dynamics in the protochordate Ciona intestinalis reveals how neuromodulators flexibly shape its behavioral repertoire"

We expect to receive your revised manuscript within two weeks. 

*Published Peer Review History*

*Press*

Sincerely,

Kris

Kris Dickson, Ph.D. (she/her)

Neurosciences Senior Editor/Section Manager,

kdickson@plos.org,

PLOS Biology

DATA POLICY:

Note that we do not require all raw data. Rather, we ask that all individual quantitative observations that underlie the data summarized in the figures and results of your paper be made available either as supplemental files or via deposition in a publicly available database with an open access code. 

The Supplementary files (e.g., excel) you provided, and have referenced in your figure legends, fit this requirement. However, we will note that you indicate some of the data are in S49 and S50 table (for Fig3 = drug stats), but you have not provided these tables. Please also note that the numerical data provided should include all replicates AND the way in which the plotted mean and errors were derived (it should not present only the mean/average values).

Also, we note that you've stated that the Ciona tierpsy derived skeleton files are also available from the Zenodo database (https://zenodo.org/record/3926785#.YZt1u7oo9yE). When we went to access this site, a message popped up saying “This site can’t be reached”

Please also ensure that figure legends in your manuscript include the correct information on where the underlying data can be found, and ensure your supplemental data file/s has a legend.

DATA NOT SHOWN?

Reviewer remarks: N/A

---

## [Editor Report · Decision Letter 3]

6 Jul 2022

Dear Dr Chatzigeorgiou,

Thank you for the submission of your revised Research Article "Comprehensive analysis of locomotion dynamics in the protochordate Ciona intestinalis reveals how neuromodulators flexibly shape its behavioral repertoire" for publication in PLOS Biology, and for comprehensively addressing our editorial requests. On behalf of my colleagues and the Academic Editor, Piali Sengupta, I am pleased to say that we can in principle accept your manuscript for publication. 

At this stage, we simply need you to address any remaining formatting and reporting issues that will be detailed in an email you should receive within 2-3 business days from our colleagues in the journal operations team. No action is required from you until then. Please note that we will not be able to formally accept your manuscript and schedule it for publication until you have completed any of their requested changes.

PRESS

We frequently collaborate with press offices. If your institution or institutions have a press office, please notify them about your upcoming paper at this point, to enable them to help maximize its impact. If the press office is planning to promote your findings, we would be grateful if they could coordinate with biologypress@plos.org. If you have previously opted in to the early version process, we ask that you notify us immediately of any press plans so that we may opt out on your behalf.

Sincerely, 

Kris

Kris Dickson, Ph.D. (she/her)

Neurosciences Senior Editor/Section Manager

PLOS Biology

kdickson@plos.org